# Guiding cell adhesion and motility by modulating cross-linking and topographic properties of microgel arrays

**Janine Riegert[1]ⓒ, Alexander Töpel[2,3]ⓒ, Jana Schieren[1], Renee Coryn[1], Stella Dibenedetto[1], Dominik Braunmiller[2,3], Kamil Zajt[1], Carmen Schalla[1], Stephan Rütten[4], Martin Zenke[1], Andrij Pich[2,3], Antonio Sechi[1]***

**1** Dept. of Cell Biology, Institute of Biomedical Engineering, RWTH Aachen University, Aachen, Germany,
**2** Functional and Interactive Polymers, Institute of Technical and Macromolecular Chemistry, RWTH Aachen University, Aachen, Germany, **3** DWI, Leibniz Institute for Interactive Materials e.V., Aachen, Germany,
**4** Electron Microscopy Facility, Institute of Pathology, RWTH Aachen University, Aachen, Germany

ⓒ These authors contributed equally to this work.
* antonio.sechi@rwth-aachen.de

**Data Availability Statement:** All relevant data are within the paper and its Supporting Information files.

## Abstract

Biomaterial-driven modulation of cell adhesion and migration is a challenging aspect of tissue engineering. Here, we investigated the impact of surface-bound microgel arrays with variable geometry and adjustable cross-linking properties on cell adhesion and migration. We show that cell migration is inversely correlated with microgel array spacing, whereas directionality increases as array spacing increases. Focal adhesion dynamics is also modulated by microgel topography resulting in less dynamic focal adhesions on surface-bound microgels. Microgels also modulate the motility and adhesion of Sertoli cells used as a model for cell migration and adhesion. Both focal adhesion dynamics and speed are reduced on microgels. Interestingly, Gas2L1, a component of the cytoskeleton that mediates the interaction between microtubules and microfilaments, is dispensable for the regulation of cell adhesion and migration on microgels. Finally, increasing microgel cross-linking causes a clear reduction of focal adhesion turnover in Sertoli cells. These findings not only show that spacing and rigidity of surface-grafted microgels arrays can be effectively used to modulate cell adhesion and motility of diverse cellular systems, but they also form the basis for future developments in the fields of medicine and tissue engineering.

## Introduction

Biomaterials are often use as guidance structures in a variety of applications. For instance, biomaterials can be used to deliver pharmaceutically active compounds or cells to specific locations and can contribute to the repair of damaged tissues. Furthermore, biomaterials can mimic the physical and chemical features of the extracellular matrix thus supporting wound healing [1–4]. At the cellular level, biomaterial chemistry and topography are often exploited to regulate numerous cellular processes including differentiation, cell adhesion and migration as well as dendritic cell function [5–11].

**Funding:** This work was partly supported by the Center for Chemical Polymer Technology (CPT), which was supported by the EU and the federal state of North Rhine Westphalia (grant EFRE 30 00883 02). AP thanks the financial support of the Deutsche Forschungsgemeinschaft (DFG) of the Collaborative Research Center SFB 985 "Functional Microgels and Microgel Systems". The funders have no role in study design, data collection and analysis, decision to publish, or preparation of the manuscript. There was no additional external funding received for this study.

**Competing interests:** The authors have declared that no competing interests exist.

Microgels play a central role in several aspects of the biomaterials research. Microgels are colloids characterized by distinctive physical and chemical properties, which include a porous structure, swelling in aqueous media, surface activity, and a very flexible chemical functionality. Another fundamental feature of microgels is their high responsiveness to several external stimuli such as temperature, pH, light, redox potential, magnetic fields and enzymes [12–16]. All of these features make microgels crucial building blocks in the context of several applications such as coatings, drug and gene-delivery systems, catalysis, water purification, sensing devices and cosmetic applications [17–25]. In addition, microgels can be readily attached to solid substrates (physically or chemically) to form linear arrays or films [26]. In this context, we have developed a printing technology that allows controlled functionalization of solid substrates with microgels and the variation of several microgel properties, for instance topology, degree of swelling and chemical structure [27].

In the context of biological applications, microgels have been used for drug and nucleic acid delivery as well as tissue regeneration [28–33]. Moreover, temperature-responsive microgels have been specifically targeted to cancer cells to induce their necrosis or apoptosis [34,35]. Inhibition of tumor cell proliferation has also been achieved via the release of doxorubicin or paclitaxel from pH-sensitive microgels [36,37]. Finally, advanced tissue engineering applications have included the support of mouse fibroblasts cell adhesion and proliferation [38] and the regulation of the adhesion of different cell types using temperature-sensitive microgels [39].

Cell adhesion and migration are two fundamental biological processes required for biomaterial-supported tissue regeneration and engineering. Hence, the need for tailored interfaces and guidance systems that mimic the extracellular matrix thus supporting cell adhesion and migration. Several studies have proposed a number of strategies for controlling cell adhesion and migration by biomaterials. To mention a few, linear random cell migration has been promoted using microgrooves [40–42]. Moreover, complex geometries such as asymmetric teardrop islands have been shown to be able to convert random cell migration to directional cell migration [43–46]. Cell adhesion of various cell types has also been regulated by the use of elliptical rings with tunable height and shape or polymer brush nanoarrays [47–49]. In a previous study, we have demonstrated that microgels can be used to control cell adhesion and migration [50]. Substrate-anchored microgel arrays greatly influenced the distribution and orientation of focal adhesions and the actin cytoskeleton resulting in the alignment of these cytoskeletal structures in parallel with the microgel arrays. Remarkably, increasing the spacing of the microgel arrays from 1000 to 2000 nm augmented the motility of B16F1 cells by a factor of 2. In addition, temperature-responsive reduction of microgel stiffness has been shown to effectively regulate cell migration [50]. These findings demonstrate that microgels can be used not only for investigating important aspects of cell migration, but also for supporting and tailoring such behavior.

To refine the understanding of the impact of surface-grafted microgel arrays on cell adhesion and migration, it is necessary to expand the range of topographic and mechanical features of microgel arrays and test their effect on different cell types. To this end, we generated a set of microgel arrays in which (i) the spacing between adjacent arrays was varied between 300 and 1600 nm, or (ii) their stiffness was varied by changing their degree of cross-linking (2.3 or 5 mol%). We studied the influence of these microgel arrays features on cell adhesion and migration using two model cell types: melanoma and Sertoli cells. Furthermore, we tested whether Gas2L1 (growth arrest specific 2 like 1), a target of thyroid hormone receptor that is associated with the actin and microtubule cytoskeletons and is also important for focal adhesion dynamics and cell migration [51–53], plays a role in the adhesion and migration of Sertoli cells on microgel arrays. The present findings clearly show that spacing and rigidity of surface-grafted

microgel arrays can be manipulated to effectively modulate cell adhesion and motility of diverse cell types.

## Materials and methods

### Materials

*N*-Isopropylacrylamide (NIPAm, Acros Organics 99%) was recrystallized from hexane before use. *N,N'*-Methylenebis(acrylamide) (BIS, Sigma-Aldrich, 99%) and 2,2'-Azobis(2-methylpropionamidine)-dihydro-chloride (AMPA, Sigma-Aldrich, 97%) and hexadecyl(trimethyl) ammonium bromide (CTAB, Sigma-Aldrich 98%) were used as received.

### Microgels synthesis

Microgels were synthesized by precipitation polymerization.[54] NIPAm, BIS and CTAB were dissolved in ultra-pure water (150 mL) in a double wall reactor and heated to 70˚C (see S1 Table in S1 Appendix for more details). Nitrogen was purged over the solution for 30 minutes. AMPA was dissolved in a small amount of water and added to initiate the polymerization, which lasted for 4 hours. Microgels were subsequently purified by dialysis against 5 liters of deionized water for seven days with repeated exchange of water (three times each day) using a membrane with a molecular weight cut-off (MWCO) of 12,000–14,000 Da (ZelluTrans, Roth). The concentration of the microgel solution was determined by gravimetric analysis. Microgels were stored in water, which was removed by lyophilization prior to use.

### Preparation of PDMS wrinkles

PDMS stamps were prepared as described earlier [50,55,56]. PDMS was produced from the dual component Sylgard 184 elastomer kit (Dow Chemical) by mixing the monomer (33 g) with the base (3.3 g) for one minute and pouring the solution into a 10x10 cm plate to obtain a 3 mm thick film. PDMS solution was pre-cured and degassed over night at room temperature before final curing at 80˚C for two hours. To produce the wrinkled stamps, a custom-made stretching device was used. A 1 x 2.5 cm block of PDMS was clamped into the device and stretched to 130% of its original size by increasing the distance between the clamps from 1.3 cm to 1.7 cm. Oxidation of the PDMS surface was performed in a low-pressure plasma oven (Plasma Activate Flecto 10 USB; Plasma Technology GmbH, Germany) with ambient air plasma at a pressure of 0.2 mbar and a power of 100 W. This process was performed for either 15 sec (300 nm), 120 sec (800 nm), 480 sec (1200 nm) or 900 sec (1600 nm), after which the tension was released and the wrinkled PDMS stamp was placed on a glass surface to maintain its stability.

### Printing of microgels on glass substrates

Microgels were printed on glass coverslips as described earlier [27,50]. Briefly, glass coverslips were cleaned by sequential exposure to acetone, water and isopropanol in an ultrasonic bath (5 minutes each) followed by drying in a stream of nitrogen and activation in a plasma oven (Plasma Activate Flecto 10 USB; Plasma Technology GmbH, Germany) at 0.2 mbar for 300 sec. For the printing process, 15 µL of microgel solution was placed in the middle of a glass cover slip. For printing microgel arrays, wrinkled PDMS stamps were used (see section above), whereas for printing microgel films PDMS stamps, which were not stretched before oxidation, were used. The stamp was placed on the glass coverslip at a tilted angle and gently dropped onto the microgel droplet. Air bubbles and excess microgel solution were removed by gently pushing the stamp with tweezers. The stamp/coverslips combination was allowed to dry

overnight (or at least for 12 hours). After removing the PDMS stamp, microgels were grafted to the surface by low pressure argon plasma in a plasma oven (Plasma Activate Flecto 10 USB; Plasma Technology GmbH, Germany). The oven was purged five times, by changing the pressure between 0.5 mbar and 0.1 mbar for cycles of 30 sec. Pressure was equilibrated for 60 sec at 0.2 mbar prior to cross-linking the microgel surface with argon plasma. Surface activation was performed for 23 sec at a pressure of 0.2 mbar and a power of 100 W. From this point, microgels could be used immediately or stored at room temperature.

## Characterization of microgels and microgel arrays

The hydrodynamic diameter ($D_{H,x°C}$) of microgels was determined by dynamic light scattering (DLS). For this purpose, 5 µL of microgel solution was diluted with 1.2 mL of ultra-pure water and measured with a Zetasizer ZS (Malvern Instruments GmbH) using a 633 nm laser and analyzing its back scatter at 173°. To investigate the thermoresponsive properties of microgels, the hydrodynamic diameter was measured at temperatures between 15°C to 50°C in 1°C steps. The volume phase transition temperature (VPTT) was determined as the inflection point in the plot of the hydrodynamic radius versus the temperature. The degree of microgel swelling ($R_{x°C,\ 50°C}$) was calculated by comparing the hydrodynamic diameter at x°C to the hydrodynamic diameter in the collapsed state ($D_{H,\ 50°C}$) according to the Eq (1):

$$R_{x°C/50°C} = \frac{V_{x°C}}{V_{50°C}} = \frac{\frac{4}{3} \cdot \pi \cdot \left(\frac{D_{H,x°C}}{2}\right)^3}{\frac{4}{3} \cdot \pi \cdot \left(\frac{D_{H,50°C}}{2}\right)^3} = \left(\frac{D_{H,x°C}}{D_{H,50°C}}\right)^3 \qquad 1$$

To investigate the structure of the microgel surface, atomic force microscopy (AFM) measurements were performed. To this end, 1x1 cm silica wafers were cleaned for 15 minutes in toluene in an ultrasonic bath and dried in a nitrogen stream. The dried wafers were further cleaned with a high-pressure carbon dioxide jet stream. The cleaned silica wafers were activated for 300 sec in a low-pressure plasma oven at 0.2 mbar (Plasma Activate Flecto 10 USB; Plasma Technology GmbH, Germany). A volume of 50 µL of a 1% diluted microgel solution was spin coated onto the activated wafer (WS-650-SZ-6NPP/Lite, Laurell) at an acceleration of 800 rpm/s and a speed of 2000 rpm for 1 minute. For AFM analyses, a NanoScope V (Digital Instruments Veeco Instruments Santa Barbara, CA) equipped with a J-Scanner was used. Uncoated NCH-50 (Nano World Point probe) cantilevers were used as probes with a resonance frequency of 320 kHz and a force constant of 42 N m$^{-1}$. All measurements were performed in tapping mode and the images were analyzed with Gwyddion (version 2.53). The contact diameter of the microgels ($D_{AFM}$) and the height of the microgels ($h_{AFM}$) in dry state could then be determined from the AFM values. The deformation of microgels was calculated using the Eq (2):

$$\text{deformation} = \frac{D_{AFM}}{h_{AFM}} \qquad 2$$

The stability of microgel arrays in aqueous solutions was determined by placing them in ultra-pure water for up to 48 hours. Samples were then left to dry under ambient conditions and analyzed by AFM as described above.

## Cell culture

B16F1 cells (ATCC CRL 6323) and B16F1 cells stably expressing RFP-zyxin [10,50] were grown in DMEM high glucose supplemented with 10% FCS, 2 mM L-glutamine, 1 mM

sodium pyruvate, 100 µg mL$^{-1}$ streptomycin and 100 U mL$^{-1}$ penicillin at 37˚C, 5% CO$_2$. Control and Gas2L1 knock out Sertoli cells were grown in DMEM/F12 [1:1] supplemented with 10% FCS, 2 mM L-glutamine, 100 µg mL$^{-1}$ streptomycin and 100 U mL$^{-1}$ penicillin at 37˚C, 5% CO$_2$ [51].

## Immunofluorescence and scanning electron microscopy

Cells were fixed and permeabilised as described earlier [10,50,57]. Briefly, the actin cytoskeleton was labelled with Alexa 594-conjugated phalloidin (0.3 U mL$^{-1}$, cat. no. A12381, Thermo Fischer). Nuclei were labelled with the DAPI (5 µg mL$^{-1}$, cat. no. D1306, Thermo Fischer). Vinculin was labelled using an anti-vinculin antibody (1:400, cat. no. V9131, hVin1, Sigma-Aldrich) followed by Alexa 594-conjugated goat anti-mouse IgG (2 µg mL$^{-1}$, cat. no. A11005, Thermo Fischer). Cover slips were mounted in Prolong Gold antifade agent (cat. no. P36934, Thermo Fischer). Images were acquired with a cooled, back-illuminated charge-coupled device camera (Cascade 512B; Princeton Instruments, USA) driven by IPLab Spectrum software (Scanalytics, USA) using a Plan-Apochromat 100x/1.30 numerical aperture oil immersion objective. Scanning electron microscopy was performed as described earlier [10,50,51,57].

## Imaging and analysis of cell motility and focal adhesion dynamics

To analyze cell motility, cells seeded on glass cover slips, microgel films or arrays were imaged for 24 h (at 37˚C and 5% CO$_2$) using an Axio Observer Z1 inverted microscope (Carl Zeiss, Germany) equipped with a Plan-Apochromat 10x objective and an AxioCam MRm (Carl Zeiss, Germany) driven by Zen 2 software (Carl Zeiss, Germany). Images were acquired every 5 min at multiple locations using a motorized X-Y stage. To determine the average speed and directionality of cell motility, manual tracking of the cells' centroid was done using the ImageJ plugin MTrackJ [58]. Directionality of cell movement was calculated by analyzing all angular displacements measured between subsequent frames as described earlier [50].

Imaging of focal adhesion dynamics was performed by total internal reflection fluorescence (TIRF) microscopy using an Axio Observer Z1 inverted microscope equipped with a motorized TIRF slider (Carl Zeiss, Germany). Excitation of RFP-zyxin was carried out using a 561 nm laser (running at 10% of the nominal output power of 100 mW). The depth of the evanescent field was ≈70 nm. Images were acquired every 10–15 sec using an Evolve Delta EM-CCD camera driven by ZEN 2 software (Carl Zeiss, Germany). For all experiments, exposure time, depth of the evanescent field, and electronic gain of the EM-CCD camera were kept constant. The analysis of focal adhesion dynamics was achieved using a segmentation and tracking algorithm [59,60] to determine the following focal adhesion parameters: assembly and disassembly rates, area, life span and speed (i.e., speed of the apparent movement of FAs relative to the substrate).

To determine the turnover of zyxin within focal adhesions, Sertoli and B16F1 cells stably expressing RFP-zyxin were used [51,57]. Briefly, focal adhesions were imaged by TIRF and fluorescence recovery after photobleaching (FRAP) microscopy for 15–20 min. One min after the beginning of image acquisition, a portion of a single focal adhesion ( 1 µm) was bleached for 1 second using a 405 nm laser at maximum power (100 mW) driven by a UGA-40 control unit (Rapp Opto Electronic GmbH, Germany). The same conditions (area bleached and the duration and intensity of the laser impulse) were applied for all experiments [51,61]. FRAP analysis was performed in two steps. Firstly, ImageJ (developed by Rasband, W.S., National Institute of Health, Bethesda, USA, http://imagej.nih.gov/ij/) was used to measure the average pixel intensity of three distinct regions of interest (ROI): ROI1: bleached area; ROI2:

unbleached area within the cell; ROI3: background. Secondly, easyFRAP was used to normalize the FRAP recovery curves and calculate the mobile fractions as described [62].

## Statistical analysis

150–200 samples were analyzed (i.e., motile cells or dynamic focal adhesions) from 2–3 independent experiments. For the motility studies, cellular speed and directionality were analyzed, whereas for focal adhesion dynamics studies, assembly and disassembly rates, speed, size and life span of focal adhesions were analyzed. For the analysis of zyxin turnover at focal adhesions, its mobile fraction was analyzed. Prism 8 (GraphPad Software Inc., CA) was used to generate all graphs and statistics. Pairwise statistical analyses were performed using the two-tailed Mann–Whitney nonparametric *U*-test and the null hypothesis (the two groups have the same median values, i.e., they are not different) was rejected when $p > 0.05$. Multiple comparisons were performed using the one-way ANOVA test in combination with the Tukey method with a statistically significant difference set at $p < 0.05$. In all box plots, the line in the middle of the box indicates the median, the top of the box indicates the 75th quartile, whereas the bottom of the box indicates the 25th quartile. Whiskers represent the 10th (lower) and 90th (upper) percentile, respectively.

## Results

### Preparation and characterization of microgels

We have previously demonstrated that cell migration can be effectively modulated by changing the spacing and the degree of microgel array swelling [50]. Hence, we decided to analyze these aspects in more detail by generating microgel arrays using a higher amount of cross-linker or by varying their spacing from 300 to 1600 nm.

Since the generation of microgel arrays with smaller spacing requires microgels with a small hydrodynamic diameter, we initially concentrated our efforts on setting up a method that would readily allow the control of this parameter. To this end, we took advantage of surfactants, which are known to stabilize precursor microgel particles during the polymerization process, resulting in smaller microgel particles [63–65]. Specifically, we synthesized *N*-Isopropylacrylamide (NIPAm) microgels, cross-linked by *N*,*N*-Methylenebis(acrylamide) (BIS), in the presence of the surfactant hexadecyl(trimethyl)ammonium bromide (CTAB) at the concentration varying between 0wt% and 2.5wt% of the total mass of all products, keeping the concentrations of monomer, cross-linker and initiator constant (S1 Table in S1 Appendix). For simplicity, we will refer to the four microgel preparations as: MG small (generated in the presence of 2.5mol% CTAB), MG medium (0.5mol% CTAB), MG large (0mol% CTAB) and MG large-stiff (0mol% CTAB, 5mol% BIS). The hydrodynamic diameters of microgels were determined by dynamic light scattering (DLS) (Table 1).

**Table 1. Physical properties of microgels.**

|  | $D_{H20°C}$ (nm) | $D_{H50°C}$ (nm) | $R_x$ (15°C/50°C) (a.u.) | VPTT (°C) |
|---|---|---|---|---|
| MG large-stiff | 740±18 | 384±1 | 8.44 | 33.1 |
| MG large | 753±22 | 348±2 | 10.76 | 32.2 |
| MG medium | 555±16 | 257±3 | 10.91 | 32.4 |
| MG small | 162±2 | 67±1 | 16.34 | 32.4 |

$D_H$: Hydrodynamic diameter; $R_x$: Swelling degree; VPTT: Volume phase transition temperature.

The hydrodynamic diameter of the microgels in the swollen state (20°C), at temperatures below the volume phase transition temperature (VPTT), could be decreased by increasing the amount of CTAB from 753 nm to 162 nm. At the typical temperature of a cell culture (37°C), clearly above the VPTT, the water was released from the polymer network and the hydrodynamic diameter decreased to values between 348 nm and 67 nm. In addition, the degree of swelling and chemical structure of the microgels was almost unaffected by CTAB (S1 Fig in S1 Appendix). It is important to note that the temperature responsive properties are essential for the synthesis of the microgels but were not used as a trigger for modulating cell behavior in this study.

The Raman (S2A Fig in S1 Appendix) and FTIR (S2B Fig in S1 Appendix) spectra indicated that the chemical structure of microgels was also unaffected by the addition of CTAB during the synthesis and that the samples were free of surfactant after purification. Importantly, since CTAB is known to be cytotoxic [66], microgel preparations were extensively dialyzed to completely remove CTAB, as indicated by the Raman spectra (S2A Fig in S1 Appendix).

Atomic force microscopy (AFM) images were taken to investigate microgel morphology. All microgels had a rounded shape (Fig 1A–1D) and the diameter in dry state ($D_{AFM}$) decreased with increasing amount of surfactant that was used. Moreover, the contact area of the microgel with the surface and their height above the surface decreased with increasing amount of CTAB, whereas the deformation of the microgel was rather unaffected (S2 Table in S1 Appendix, S2 Fig in S1 Appendix). It must be noted that the size of the smallest microgels was in the range of the limits of the measurement method, thus causing a large variation in the measurements. Furthermore, smaller microgels have a more homogeneous structure of the polymer network, which lead to higher spreading [63]. Taken together, these observations show that the use of CTAB during the reaction does not grossly alter the final microgel properties and is an easy way to control the size of the microgel.

To increase microgel stiffness, cross-linker concentration was changed from 2.35mol% to 5mol%, without altering the concentration of any other component (S1 Table in S1 Appendix). The physico-chemical characterization showed that large stiff microgels were comparable in many aspects to the large microgels used previously by our group [50] including temperature responsiveness, hydrodynamic diameter in swollen state and chemical composition (Fig 1, S1 Fig in S1 Appendix, S1 Table in S1 Appendix). Moreover, the higher cross-linker concentration resulted in microgels having a greater height and smaller contact area in the dry state (S2 Fig in S1 Appendix), thus leading to lower deformability (S2 Table in S1 Appendix).

## Printing and characterization of surface-bound microgel arrays

To print microgel arrays with different spacing, we adopted a previously published approach [27,50]. However, since the present goal was to fabricate arrays with smaller spacing, it was necessary to consider two fundamental parameters: (i) the size of the microgel and (ii) the wavelength of the PDMS stamp, defined by the thickness of its oxidized surface (i.e., the time of plasma activation) (S4 Fig in S1 Appendix). These two parameters have to complement each other, since any mismatch (e.g., large microgels used to generate smaller spacing), would results in "crippled" microgel arrays. As described earlier, all arrays were cross-linked by low pressure argon plasma to enhance their stability in aqueous media [50]. The argon plasma creates radicals in the polymer chains, which recombine forming a covalent bond [67,68].

Hiltl and colleagues [27] found that optimal alignment of microgels with different chemical composition could be achieved with a ratio of the wavelength of the PDMS stamp to the hydrodynamic diameter of the microgels of $D_H$x1.2< λ<$D_H$x2.0. By combining pNIPAm-based microgels with different hydrodynamic diameters and PDMS stamps of different wavelength,

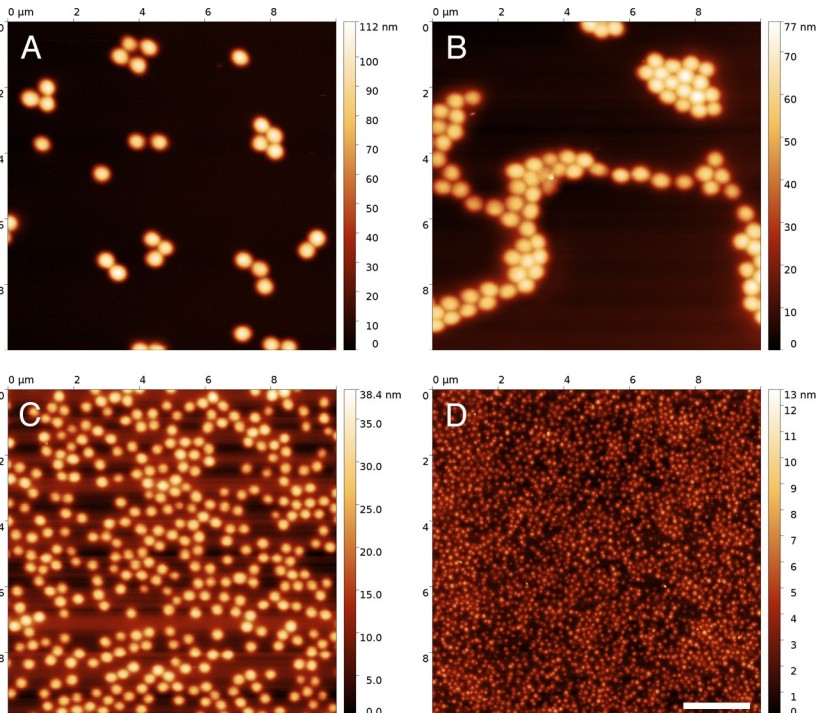

**Fig 1. Analysis of microgel physical and chemical properties.** (A-D) Atomic force microscopy images showing microgels generated using 5mol% cross-linker (A) or 0% (B), 0.5% (C) or 2.5% (D) CTAB. The legend on the side of each atomic force microscopy image indicates the height of the arrays. Scale bar: 2 μm.

the ideal ratios between microgel hydrodynamic diameter and stamp wavelength for the present system were determined to be $1.5xD_{H,20°C} < \lambda < 2.5xD_{H,20°C}$. Accordingly, the microgels synthesized in the absence of CTAB were chosen for the 1600 and 1200 nm arrays, with a ratio of $\lambda \approx 1.7xD_{H,20°C}$ and $\lambda \approx 2.2xD_{H,20°C}$, respectively. For the 800 nm and 300 nm arrays, microgels synthesized with 0.5% ($\lambda \approx 1.9xD_H$) and 2.5% CTAB ($\lambda \approx 1.7xD_H$) were chosen. The plasma activation times for the PDMS stamp wavelengths of 300, 800, 1200 or 1600 nm were 15, 120, 480 and 900 seconds, respectively. Stiffer microgel arrays (synthesized with 5mol% BIS) were printed using a standard spacing of 1200 nm.

AFM analysis showed that the printed arrays in the dry state had the expected spacing and morphology, being replicas of the stamp (S3 Fig in S1 Appendix). In addition, incubation of all arrays in water for up to 48 hours did not change array morphology, directionality or spacing (Fig 2), thus demonstrating their stability under these conditions. The actual spacing of the microgels with 2.35mol% cross-linker were determined to be 1615, 1208, 843 and 371 nm by AFM. Moreover, the heights of the microgel lines in the dry state were 134, 57, 44 and 15 nm (see in S1 Appendix for details).

## Surface-grafted microgels effectively regulate morphology, actin cytoskeleton and focal adhesion organization in Sertoli cells

Surface-grafted microgels have been demonstrated to be very effective tools for regulating actin cytoskeleton architecture and the size and dynamics of focal adhesions [50]. Since these findings were based on the use of B16F1 mouse melanoma cells, we sought to determine whether the surface-grafted microgel system is suitable, and to what extent, for the regulation of focal adhesion dynamics and migration of different cell types. Furthermore, we wanted to

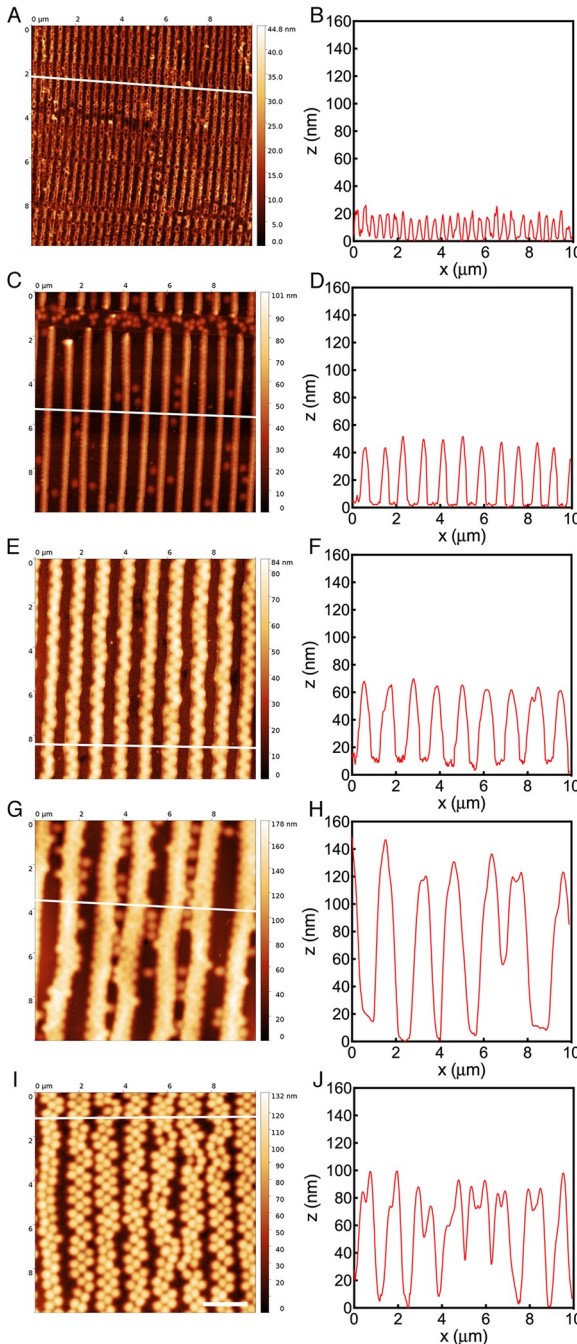

**Fig 2. Atomic force microscopy analysis of microgel array topography.** Representative atomic force microscopy images and their plot profiles of microgel arrays printed on glass coverslips at a spacing of 300 (A, B), 800 (C, D), 1200 (E, F), 1600 (G, H) nm and 5mol% (I, J) after 48 hours incubation in water. The legend on the side of each AFM image indicates the height of the arrays. White lines in the AFM images indicate the positions from which the plot profiles were generated. Scale bar: 2 μm. Note that microgel height increased by increasing array spacing and that swelling of 300 nm arrays reduced their lateral resolution causing a partial merging of adjacent arrays. The heterogeneities visible in A and C are due to the manufacturing process and do not affect the performance of the printed microgel arrays.

investigate the type of response to surface-grafted microgels of cells lacking important cyto-skeletal components.

To address these points, murine testicular Sertoli cells, which are characterized by a very well-developed actin cytoskeleton, prominent focal adhesions and a pronounced motile behavior [51] were used. Furthermore, a Sertoli knockout cell line devoid of Gas2L1, an important actin cytoskeleton-associated protein the loss of which enhances cell migration and focal adhesion turnover [51], has been engineered. These two Sertoli cell lines were, therefore, suitable for addressing the two points that were raised above.

We initially determined whether microgels had an impact on morphology and orientation of wild-type and Gas2L1 KO Sertoli cells. To this end, cells were seeded on standard microgels arrays and films that were generated using microgels synthesized with 2.35mol% cross-linker and 1200 nm spacing. Cells seeded on to glass coverslips served as controls. Both wild-type and Gas2L1 KO Sertoli cells spread efficiently on glass coverslips, often forming large lamellipodia (Fig 3A and 3D). A similar morphology and behavior could be observed in both cell lines following seeding on to microgel films (Fig 3C and 3F). It was immediately evident that both wild-type and Gas2L1 KO Sertoli cells responded to microgel arrays in a similar manner as the B16F1 cells, i.e., they acquired a marked elongated morphology and adopted an alignment parallel to the major axis of the arrays (Fig 3B and 3E).

Next, we sought to prove the association of such morphological changes with changes of the organization of focal adhesions and the actin cytoskeleton. In agreement with our previous work [51], both Sertoli cell lines that had been seeded on glass coverslips were characterized by a well-developed and prominent actin cytoskeleton (Fig 4A and 4B; upper panels). A similar arrangement of the actin cytoskeleton could also be observed when the cells were seeded on to microgel films (Fig 4A and 4B; upper panels). Given the random orientation of the actin cyto-skeleton, it was expected that also focal adhesions would also be randomly orientated in wild-

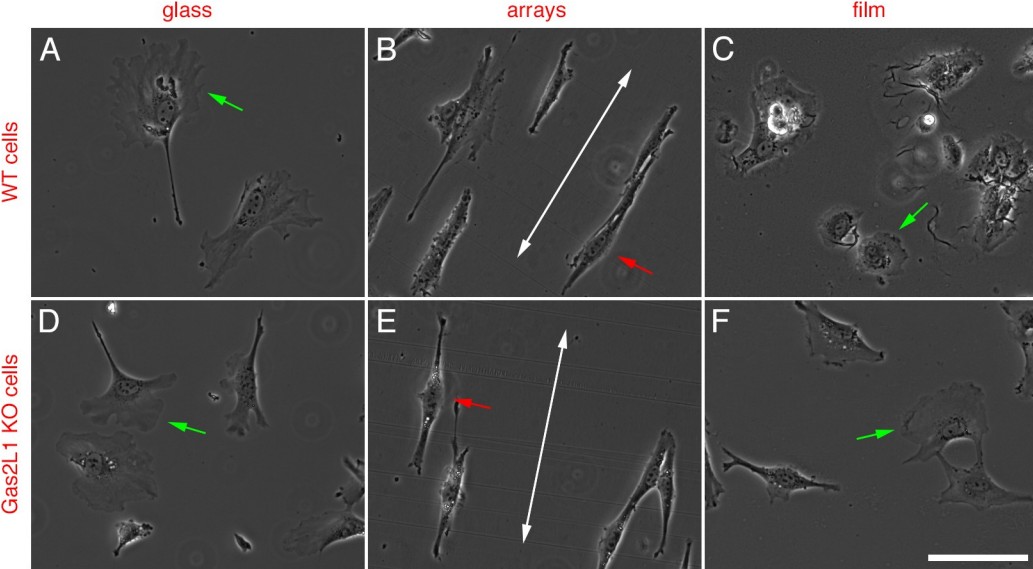

**Fig 3. Morphology of wild-type and Gas2L1 KO Sertoli cells seeded on to glass, microgel arrays and films.** Phase contrast images showing the morphology and orientation of control and Gas2L1 KO Sertoli cells on glass, microgel arrays and films. It is clear that cells on glass and microgel films demonstrate substantial spreading and the formation of lamellipodia (green arrows), whereas cells on microgel arrays are elongated with their major axis parallel to that of the array topography (red arrows). Large double headed arrows in B and E indicate the orientation of the arrays. Scale bars: 100 μm.

type and Gas2L1 KO Sertoli cells seeded on glass coverslips and microgel films (Fig 4A and 4B; green arrows in bottom panels). Consistent with the elongated cell morphology described above, the actin cytoskeleton of both Sertoli cell lines plated on microgel arrays was characterized by parallel bundles of actin filaments (possibly stress fibers) running parallel to the major axis of the arrays (Fig 4A and 4B; upper panels). According to the orientation and architecture

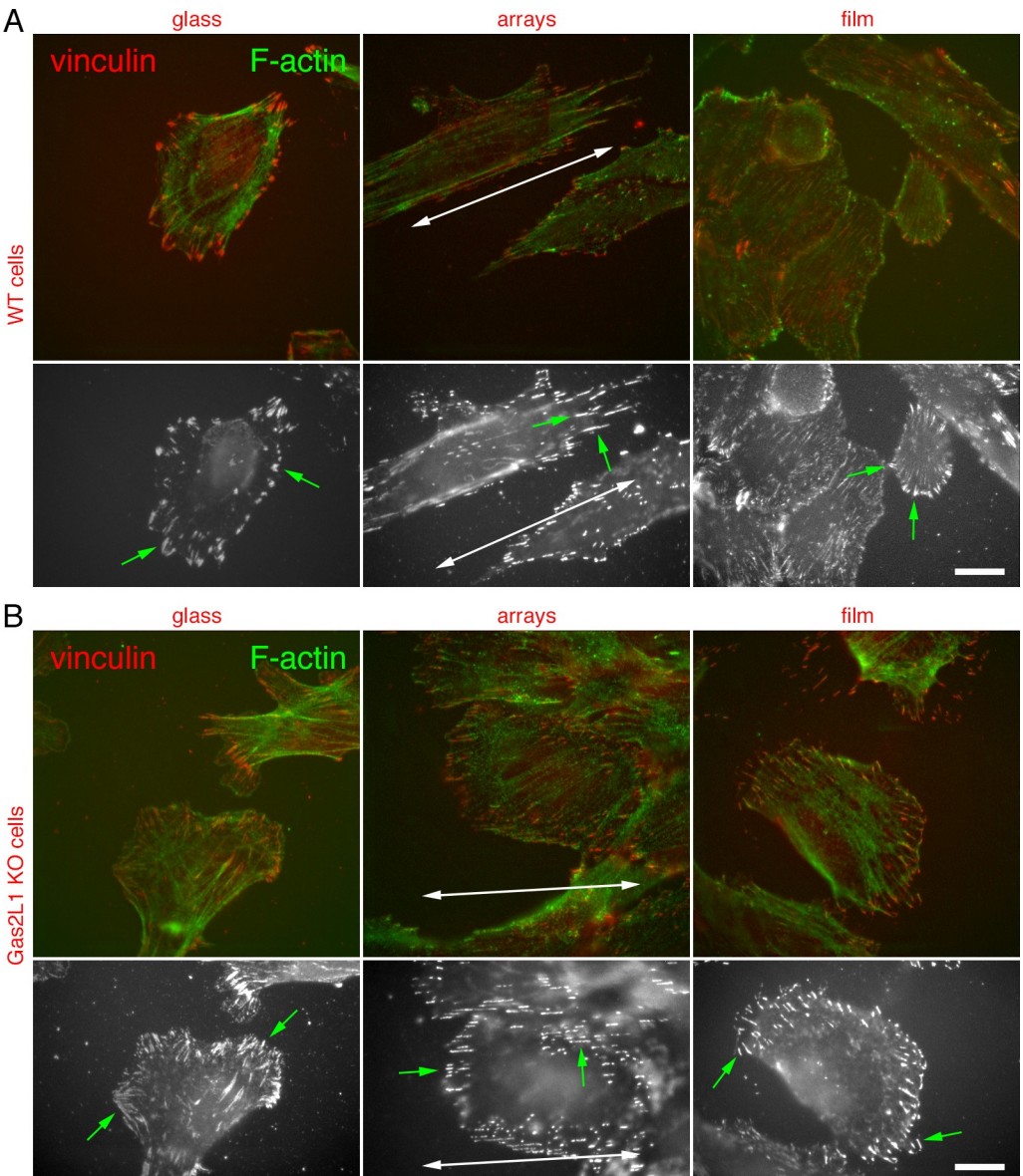

**Fig 4. Actin cytoskeleton and focal adhesion organization in wild-type and Gas2L1 KO Sertoli cells seeded on to glass, microgel arrays and films.** Control (A) and Gas2L1 KO (B) Sertoli cells were seeded on glass, microgel films and 1200 nm microgel arrays, fixed and labelled with anti-vinculin antibodies and fluorochrome-conjugated phalloidin to visualize focal adhesions and the actin cytoskeleton, respectively. Top panels in A and B show the merged vinculin and actin labelling, whereas bottom panels show only the vinculin labelling. On glass and microgel films, both control and Gas2L1 KO cells acquired a spread-out morphology and were characterized by the random orientation of their actin cytoskeleton and focal adhesions (green arrows, bottom panels in A and B). Conversely, in control and Gas2L1 cells seeded on to microgel arrays, the actin cytoskeleton and focal adhesions were orientated according to the topography of the microgel arrays. Large double headed arrows indicate the orientation of microgel arrays. Scale bars: 20 μm.

of the actin cytoskeleton, focal adhesions were also found to be elongated and orientated parallel to the major axis of the arrays (Fig 4A and 4B; green arrows in bottom panels). Collectively, these observations clearly show that the surface-grafted microgel system is effective in modulating cell morphology, actin cytoskeleton and focal adhesion organization of genotypically different populations of Sertoli cells. Furthermore, the lack of Gas2L1 in these cells does not affect their response to surface-grafted microgels.

## Surface-grafted microgel arrays modulate the migration of Sertoli cells

We have previously shown that both microgel films and arrays have been shown to reduce the migration of B16F1 cells and that microgel arrays exert a greater influence on this behavior [50]. Since surface-grafted microgels clearly influence cell shape and actin cytoskeleton and focal adhesion organization in wild-type and Gas2L1 KO Sertoli cells, we reasoned that the migration of these cells could also be modulated by surface-grafted microgels. To verify this hypothesis, the motility of wild-type and Gas2L1 KO Sertoli cells on glass coverslips, microgel films or arrays was recorded over a period of 24 hours at 37˚C, after which average speed and directionality were quantified as described earlier [50].

As shown in Fig 5, the average speed of both wild-type and Gas2L1 KO Sertoli cells on microgel arrays was significantly higher than that observed on glass coverslips. In contrast, their average speed on microgel films was significantly reduced in comparison to that observed on glass coverslips (Fig 5A and 5B). The directionality of movement for both cell types was, as expected, limited to a narrow range on the orientated microgel arrays, whereas on glass and microgel films they changed direction of movement following larger angles (Fig 5C and 5D). Thus, in analogy to our previous study [50], surface-grafted microgels can be used to effectively modulate Sertoli cell migration. It is interestingly to note that, in contrast to mouse melanoma cells [50], microgel arrays promoted the speed of migration of Sertoli cells.

Since we have previously demonstrated that Gas2L1 is important for the regulation of Sertoli cell migration and that its deletion increases the motility of these cells [51], we further reasoned that the present microgel system could be used to preferentially modulate the migration of wild-type or Gas2L1 KO Sertoli cells. A correct hypothesis would result in a clear difference in the rate of motility between the two Sertoli cell lines with the Gas2L1 cells migrating faster than wild-type cells. To address this hypothesis, pairwise comparisons (i.e., wild type vs. Gas2L1 KO) of the average speed and directionality of Sertoli cells on glass coverslips, microgel films and microgel arrays were performed. According and in support of our previous investigations [51], Gas2L1 KO Sertoli cells moved significantly faster than their wild-type counterparts on glass coverslips (Fig 5E). Furthermore, Gas2L1 KO cells were also faster than wild-type cells on microgel films and arrays (Fig 5G and 5I). The directionality of migration exhibited the cell types was not different on the three substrates (Fig 5F, 5H and 5J). These observations also indicate that Gas2L1 is dispensable for the response of Sertoli cells to surface-grafted microgels.

## Surface-grafted microgels modulate focal adhesion turnover in Sertoli cells

Since cell migration depends on the coordinated spatial and temporal regulation of focal adhesion turnover [69,70], we investigated the impact of surface-grafted microgels on focal adhesion turnover in wild-type and Gas2L1 KO Sertoli cells. To this end, wild-type and Gas2L1 KO Sertoli cells expressing RFP-zyxin [51], a component of focal adhesions, were visualized by TIRF microscopy. Focal adhesion turnover was analyzed using a dedicated algorithm [51,71] to quantify parameters including focal adhesion speed, assembly and disassembly rates.

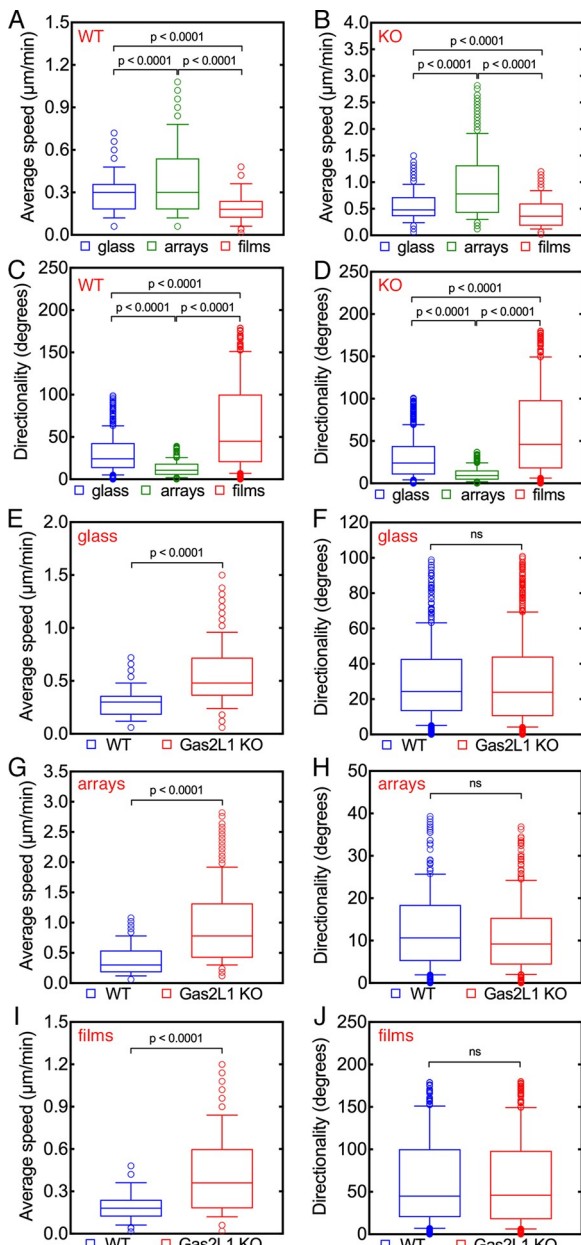

**Fig 5. Influence of microgels on cell migration and directionality of wild-type and Gas2L1 KO Sertoli cells.**
Analysis of average speed (A, B, E, G, I) and directionality (C, D, F, H, J) of control (WT) and Gas2L1 KO Sertoli cell
migration. On glass, microgel films and microgel arrays, Gas2L1 KO cells are significantly faster than control cells (A,
B, E), whereas no significant difference can be observed regarding the directionality of their migration (C, D, F). Both
control and Gas2l1 KO cells move faster on microgel arrays and slower on microgel films compared to glass controls
(A, B). Changes in the directionality of migration are significantly greater with both cell types on glass and microgel
films (H, J). Numbers indicate p values. ns: Not significant.

Focal adhesion assembly and disassembly rates were significantly reduced in both wild-type
and Gas2L1 KO Sertoli cells on microgel films compared to cells on glass coverslips (Fig 6A,
6B, 6D and 6E). Conversely, in cells on microgel arrays, focal adhesion assembly was signifi-
cantly reduced only in Gas2L1 KO cells (Fig 6A, 6B, 6D and 6E). Moreover, focal adhesion

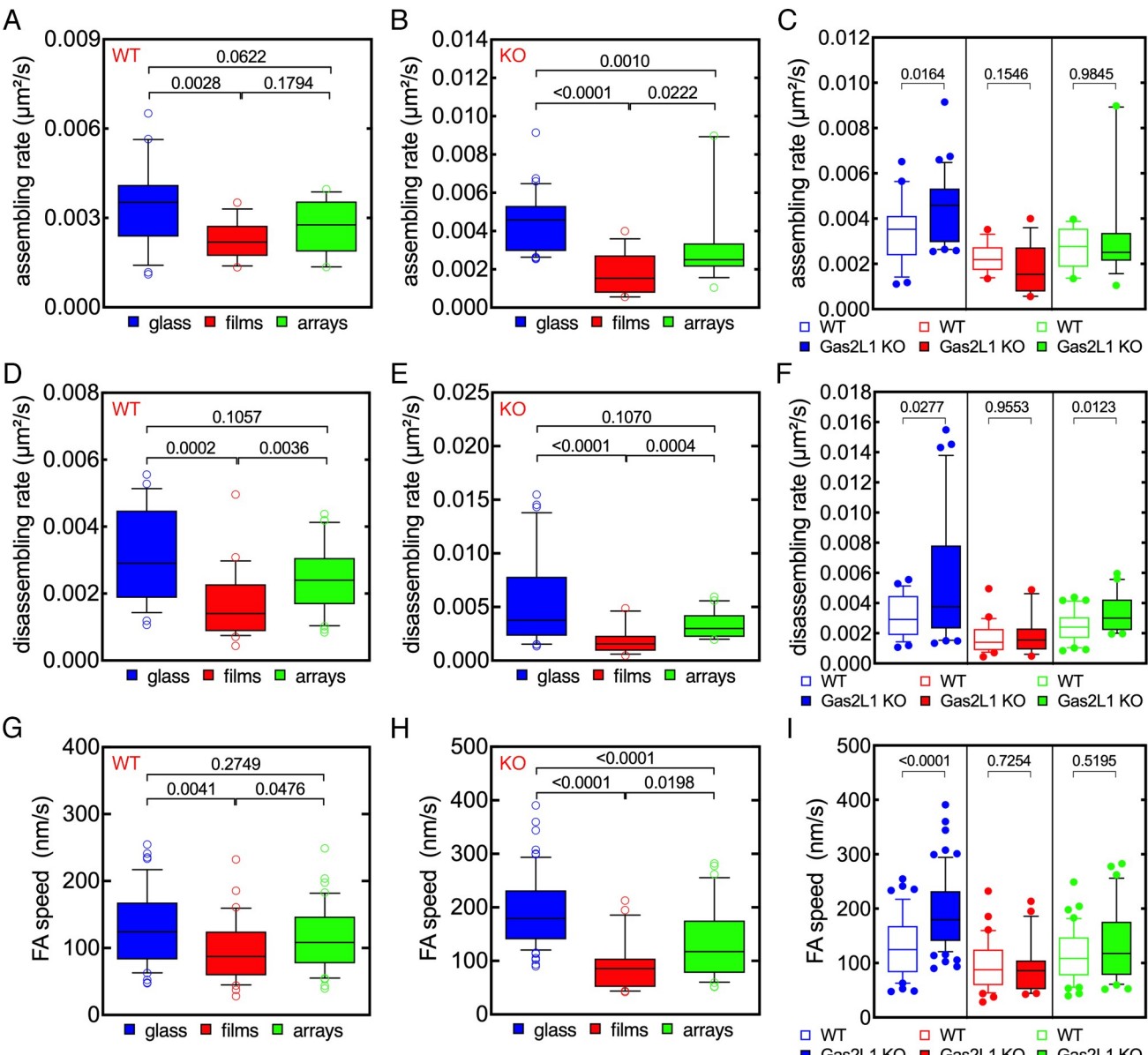

**Fig 6. Impact of microgels on focal adhesion dynamics in wild-type and Gas2L1 KO Sertoli cells.** (A-C) Control and Gas2L1 KO cells show similar FA assembly rates on microgel films and arrays. In both control (A) and Gas2L1 KO (B) cells, FA assembly rate is higher in cells seeded on to glass and lower in cells on microgel films (A, B). Pairwise comparisons show a higher focal adhesion assembly rate in cells seeded on to glass substrates. Notably, this difference is abolished when cells are seeded on to microgel films or arrays (C). (D-F) Control and Gas2L1 KO cells show similar focal adhesion disassembly rates on microgel films, but not on glass and microgel arrays. Both control (D) and Gas2L1 KO (E) cells are characterized by significantly lower levels of disassembly rates when seeded on to microgel films compared to glass and microgel arrays substrates (D, E). Pairwise analysis shows that the higher focal adhesion disassembly rate normally observed in Gas2L1 KO cells is reduced when cells are seeded on to microgel films (F). (G-I) Control and Gas2L1 KO cells show similar FA speeds on microgel films and arrays. Both control (G) and Gas2L1 KO (H) cells show the highest and lowest FA speeds on glass and microgel films, respectively (D, H). Interestingly, pairwise analysis shows that FA speed in Gas2L1 KO cells is reduced to levels comparable to those in control cells when cells are seeded on to microgel films or arrays (I). Color code for C, F and I. Blue: Glass; red: Films; green: Arrays. Numbers indicate p values. ns: Not significant.

speed was significantly lower in wild-type Sertoli cells on microgel films compared to glass coverslips, whereas focal adhesion speed was clearly reduced in Gas2L1 KO cells on microgel films and arrays (Fig 6G and 6H). It should also be mentioned that the size and life span of focal adhesions were strongly reduced in both Sertoli cell types on microgel films (S5A, S5B,

S5D and S5E Fig in S1 Appendix). Remarkably, focal adhesion size increased on Gas2L1 KO cells on microgel arrays (S5A and S5B Fig in S1 Appendix), whereas focal adhesion life span was strongly reduced in both cell types on microgel films (S5D and S5E Fig in S1 Appendix). Thus, the similar behavior of FAs in control and Gas2L1 KO cells on microgels indicates that Gas2L1 is not involved in the regulation of FA dynamics on these substrates.

Pairwise comparisons provided additional information. Specifically, on glass coverslips, focal adhesion speed, assembly and disassembly rates were higher in Gas2L1 KO than in wild-type Sertoli cells (Fig 6C, 6F and 6I). Focal adhesion speed, assembly and disassembly rates had similar magnitudes in both Sertoli cell types on microgel films and arrays (Fig 6C, 6F and 6I), however, focal adhesion disassembly rate was higher in Gas2L1 KO cells seeded on to microgel arrays. Collectively, these findings demonstrate that surface-grafted microgels can be used as an effective system to modulate focal adhesion dynamics in Sertoli cells.

## Surface-grafted microgel arrays modulate zyxin kinetics at focal adhesions in Sertoli cells

Given the robust impact of surface-grafted microgels on focal adhesion dynamics, we conducted a more detailed investigation of this behavior. For this purpose, fluorescence recovery after photobleaching (FRAP) microscopy was used to determine the kinetics of zyxin at focal adhesions. Specifically, a fixed portion of focal adhesions in wild-type and Gas2L1 KO Sertoli cells expressing RFP-zyxin was bleached with a short, high-power laser impulse and the recovery of the fluorescence signal within this area recorded over time [51,61,72,73]. As shown in Fig 7, in both wild-type and Gas2L1 KO Sertoli cells, the recovery of RFP-zyxin signal within the bleached area rapidly increased and reached a steady-state level after 200–250 seconds regardless of the substrate (Fig 7A and 7B). Notably, microgel arrays had a larger impact on RFP-zyxin recovery, causing either a reduction and an increase of zyxin kinetics in wild-type and Gas2L1 KO Sertoli cells, respectively (Fig 7A and 7B). These findings were corroborated by the analysis of the mobile fraction of RFP-zyxin, i.e., the fraction of RFP-zyxin molecules freely moving within the bleached area, showing that the recovery of RFP-zyxin in Gas2L1 KO cells on microgel arrays was significantly increased (Fig 7C and 7D). Thus, microgel arrays have a larger impact on zyxin kinetics in Gas2L1 KO cells.

## Microgel array spacing efficiently regulates cell adhesion and migration

To study the influence of diverse arrays spacing on cell migration, we chose the highly motile B16F1 cells following the reasoning that any changes of cell motility rate induced by different microgel array spacing would be more precisely detected using highly motile cells (in contrast, Sertoli cells and comparable fibroblast-like cell types typically acquire a large, flattened morphology and move less efficiently). Before analyzing the impact of diverse arrays spacing on the migration of B16F1 cells, it was necessary to determine whether cells interacted with, and responded to, the newly designed arrays. To this end, cells were seeded on 300, 800 or 1600 nm microgel arrays, incubated at 37˚C for 24 hours, and then fixed and processed for scanning electron microscopy. It was immediately evident that cells responded to the array topography, in that they acquired an elongated morphology and orientated with their major axis in parallel to the major array axis (Fig 8C and 8E). Furthermore, higher magnification images clearly showed that single cells made direct contact with microgels via cellular extensions (Fig 8D and 8F).

Next, we investigated the migration of individual B16F1 cells to determine their average speed and directionality of migration. Experimental controls included cell migration on glass coverslips and microgel films [50]. As expected, B16F1 cells moved significantly faster on glass

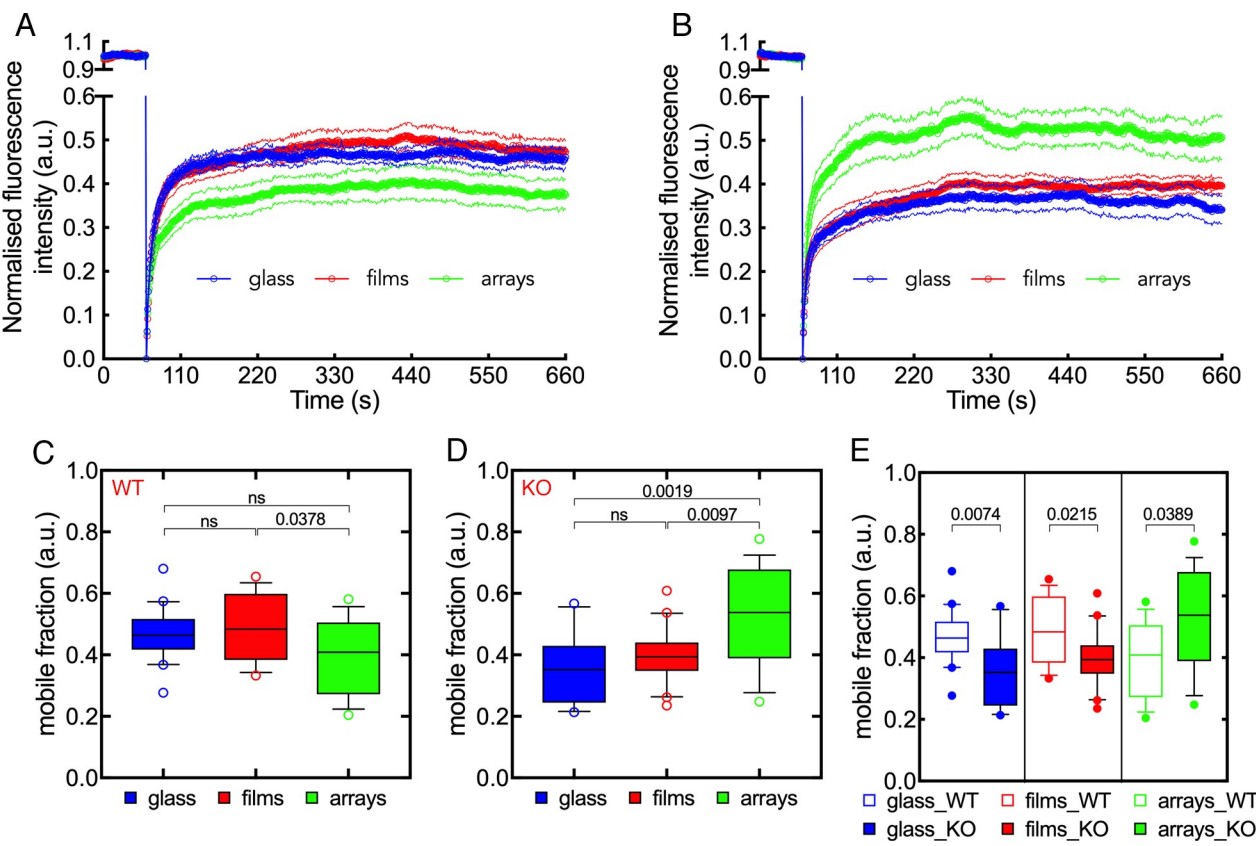

**Fig 7. Effect of microgels on zyxin kinetics in wild-type and Gas2L1 KO Sertoli.** (A, B) Fluorescence recovery after photobleaching showing zyxin kinetics at focal adhesions in control (A) and Gas2L1 KO (B) Sertoli cells on glass, microgel films and arrays. The lower magnitude of zyxin kinetics in control cells seeded on to microgel arrays is clear when compared to glass and microgel films substrates. In Gas2L1 KO cells, however, zyxin kinetics is higher on microgel arrays than on glass and microgel films. The thin lines above and below the thick curves indicate the standard error of the mean. (C-D) Comparison of the zyxin mobile fractions, showing a significantly reduced recovery in control cells on microgel arrays and a significantly elevated recovery in Gas2L1 KO cells. (E) Pairwise comparison of zyxin mobile fractions showing that, in Gas2L1 KO cells, zyxin kinetics are slower on glass and microgel films and faster on microgel arrays. Numbers indicate p values. ns: Not significant.

coverslips than on microgel films (Fig 9A, S6A Table in S1 Appendix) showing no preference in their directionality (Fig 9B, S6B Table in S1 Appendix). These observations further highlight the robustness and reliability of the surface-grafted microgel system. Regarding cells that were seeded on to microgel arrays, their average speed and directionality were greatly reduced on 800, 1200 and 1600 nm arrays (Fig 9A and 9B; S6A and S6B Table in S1 Appendix). No difference of directionality across these microgel arrays could be observed (Fig 9B). Furthermore, B16F1 migration was significantly higher on 1600 nm arrays (Fig 9A, S6A Fig in S1 Appendix). Remarkably, cells on 300 nm arrays behaved somewhat differently, in that, their migration and directionality were significantly greater compared to cells migrating over the other arrays. It should also be mentioned that cell speed was also significantly higher than their speed on microgel films (Fig 9A and 9B; S6A and S6B Table in S1 Appendix). This behavior may have been due to the formation of pseudo microgel films as a consequence of the swelling of the 300 nm arrays in cell culture media (see Fig 8).

Since this differential motile behavior most likely reflected differences in focal adhesion turnover, we quantified focal assembly and disassembly rates. As shown in Fig 9C and 9D, both assembly and disassembly rates were significantly lower in cells seeded on to microgel films, indicating a reduced focal adhesion turnover. Compared to glass control, focal adhesion

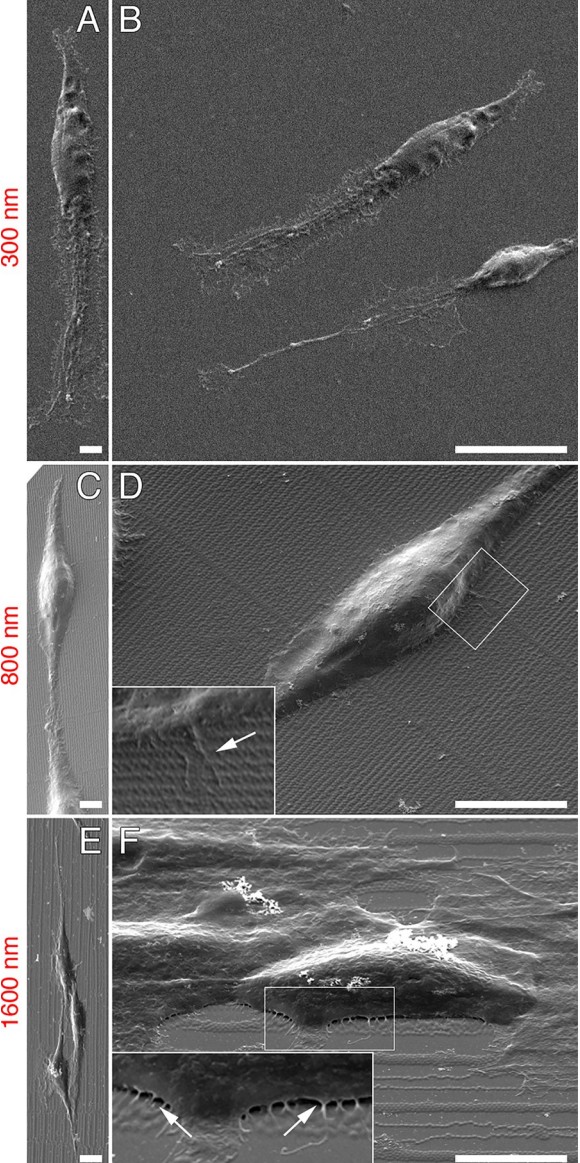

**Fig 8. Scanning electron microscopy analysis of microgel-cell interaction.** B16F1 cells were seeded on 300 nm (A, B), 800 nm (C, D) and 1600 nm (E, F) microgel arrays, incubated at 37˚C, 5% $CO_2$ for 24 hours and then fixed and processed for scanning electron microscopy. Regardless of the spacing of the array, cells fully responded to microgel topography, acquiring an elongated morphology with their major axis parallel to that of the arrays (A, C, E). Cells also generated thin projections that were in contact with microgels (arrows in insets of panels D and F). Scale bars: 4 μm (C), 10 μm (D, E, F).

formation was less effective in cells seeded on to 800 and 1600 nm arrays, as indicated by their lower assembly rate (Fig 9C; S7A Table in S1 Appendix). Similarly, focal adhesion disassembly was significantly reduced in cells on these two array variants (Fig 9D; S8B Table in S1 Appendix). It is important to note that focal adhesion turnover in cells on 300 and 1200 nm arrays was more robust than on the 800 and 1600 nm counterparts (Fig 9C and 9D; S7A and S7B Table in S1 Appendix). Remarkably, on 300 and 1200 nm arrays, focal adhesion assembly rates were not significantly different from that observed on glass control (Fig 9C; S7A Table in S1 Appendix), whereas disassembly rates were reduced (Fig 9D; S7B Table in S1 Appendix).

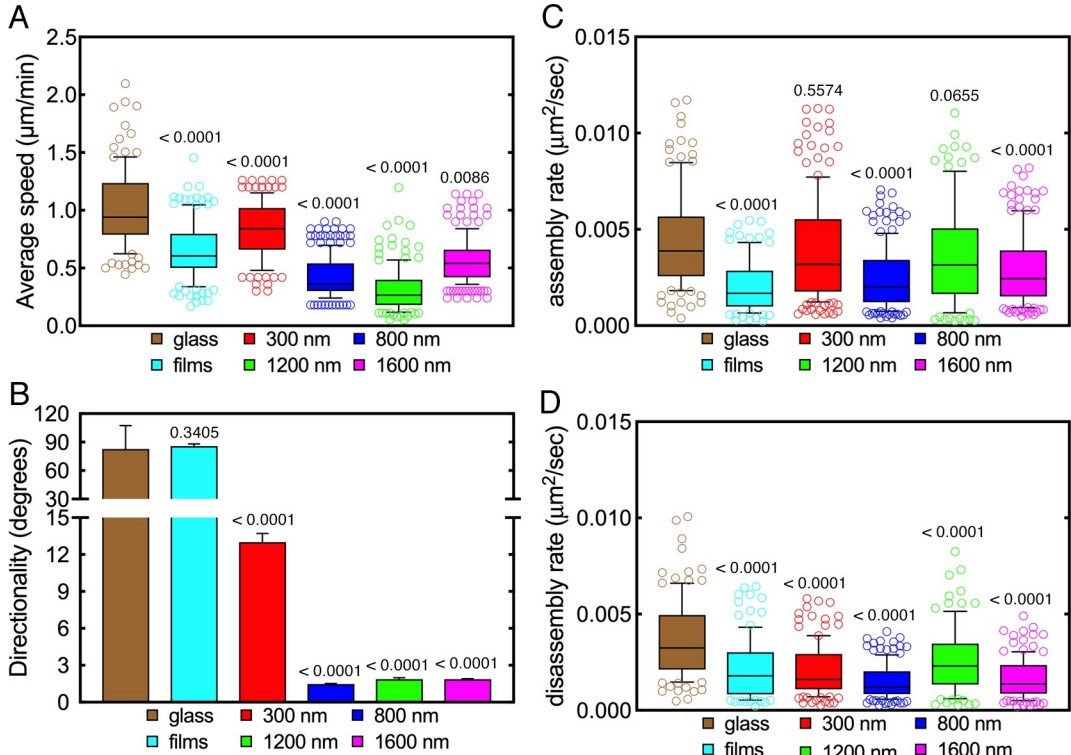

**Fig 9. Impact of the spacing of microgel arrays on cell migration and focal adhesion turnover.** (A, B) Box plots showing the impact of microgel array spacing on average speed (A) and directionality (B) of B16F1 cell migration. Compared to a control substratum (glass), B16F1 cells move significantly more slowly on microgel films. Their migration is even slower on 800 nm, 1200 nm and 1600 nm microgel arrays, but not on 300 nm microgel arrays, which support a significantly higher average speed than all other microgel substrates. Consistent with their topography, cells on microgel films display large variations in their direction of migration, similar to those on glass. The directionality of cell migration is greatly increased on 800, 1200 and 1600 nm microgel arrays, whereas cells on 300 nm microgel arrays display less directionality. (C, D) Box plots showing the impact of microgel array spacing on focal adhesion assembly (C) and disassembly (D) rates. Compared to the control substratum (glass), both focal adhesion assembly and disassembly rates on microgel films are significantly reduced. Focal adhesion assembly rate is also significantly reduced in cells seeded on to 800 and 1600 nm microgel arrays. By contrast, in cells seeded on to 300 and 1200 nm microgel arrays, focal adhesion assembly rate is not significantly different from the control. The focal adhesion disassembly rate is significantly reduced in cells on all microgel arrays. Numbers indicate p values (compared to glass). A more complete statistical analysis can be found in the in S1 Appendix.

Collectively, these findings show that microgel array spacing effectively modulates cell migration and adhesion.

## The degree of microgel array cross-linking efficiently regulates cell adhesion

To determine whether microgel stiffness could be exploited to modulate cell adhesion, we fabricated microgel arrays with different concentrations of the cross-linker (2% and 5%) to achieve soft and stiffer microgels, respectively. Since focal adhesions are widely accepted as being the most important adhesive structures in a cell and a direct proxy for cell adhesion, focal adhesion turnover was studied. Moreover, Sertoli cells were chosen due to their prominent focal adhesions, which made this analysis easier. The quantification of focal adhesion assembly and disassembly rates revealed that focal adhesion turnover was slightly, but not significantly, reduced in Gas2L1 KO cells on stiff microgel arrays when compared to wild type cells (Fig 10A). The comparison of focal adhesion turnover in wild type cells on either soft or

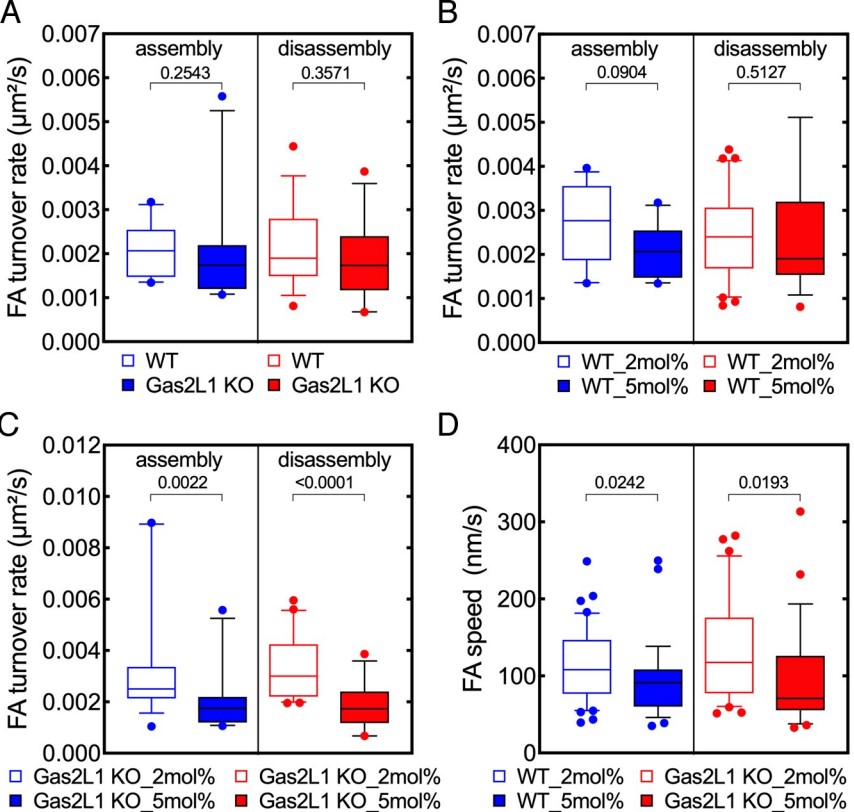

**Fig 10. Increased microgel rigidity reduces focal adhesion turnover in wild-type and Gas2L1 KO Sertoli cells.**
(A-C) Control and Gas2L1 KO Sertoli cells seeded on to soft (2mol%) or rigid (5mol%) microgel arrays. Focal adhesions were imaged by TIRF microscopy and their assembly and disassembly rates were analyzed using a dedicated algorithm. Control and Gas2L1 KO cells show similar FA assembly and disassembly rates on 5mol% microgel arrays (A). Comparison of FA turnover in control cells on soft and rigid microgel arrays shows that FA assembly rate is slightly but significantly reduced in cells on rigid microgel arrays. FA disassembly rates were not significantly different (B). Notably, in Gas2L1 KO cells, both FA assembly and disassembly rates were reduced in cells on rigid microgel arrays (C). FA speed was also significantly lower in both control and Gas2L1 KO cells seeded on to rigid microgel arrays (D). Numbers indicate p values.

stiff microgels showed no significant differences in focal adhesion assembly and disassembly rates (Fig 10B). Notably, focal adhesion assembly and disassembly rates were significantly lower in Gas2L1 KO cells seeded on to stiff microgels (Fig 10C), a clear indication of reduced focal adhesion turnover. Focal adhesion speed was also significantly reduced on stiff microgels in both cell types (Fig 10D). These observations emphasize the impact of microgel stiffness on the modulation of cell adhesion and suggest that Gas2L1 may be involved in microgel-driven regulation of cell adhesion.

## Discussion

The guidance and modulation of cellular functions such as cell adhesion and migration when associated with biomaterials is a very important and challenging task. In this study, we extended the current understanding of the impact of surface-grafted microgel arrays on cell adhesion and migration. We found that the migration of B16F1 cells is inversely correlated with microgel array spacing, whereas the directionality of their movement increased as array spacing increased. Focal adhesion dynamics in these cells was also modulated by microgels resulting in less dynamic focal adhesions on microgel arrays and films (Table 2). Surface-

**Table 2. Summary of the impact of microgels on the behavior of B16F1 cells.**

| B16F1[*] | films | 300 nm | 800 nm | 1200 nm | 1600 nm |
|---|---|---|---|---|---|
| *Average speed* | - | - | – | – | - |
| *Directionality* | n | + | ++ | ++ | ++ |
| *FA assembly rate* | – | n | – | n | – |
| *FA disassembly rate* | – | – | – | – | – |

[*]All changes refer to control cells on glass substrata. (-): Decrease; (–): Strong decrease; (+): Increase; (++): Strong increase; n: No change.

grafted microgels can also modulate the motility and adhesion of wild-type and Gas2L1 KO Sertoli cells. In these cells, focal adhesion dynamics is reduced on microgels, with films being the most effective. Interestingly, on microgel arrays, the kinetics of the focal adhesion protein zyxin was decrease in wild-type and increased in Gas2L1 KO Sertoli cells. Finally, increasing microgel stiffness caused a reduction of focal adhesion turnover (Table 3). These findings not only provide strong evidence that surface-grafted microgels are powerful tools for modulating cellular activities, but also that they form the basis for future developments in the fields of medicine and bioengineering.

To study the impact of smaller spacing of microgel arrays on cell adhesion and behavior, it was important to develop a procedure for the generation of microgels with different diameters that were tailored to the final spacing of the surface-grafted arrays yet retained the same chemical composition. In contrast to a previous study in which the PDMS stamp wavelength was chosen to fit with the microgel size [27], in this study we decided to modulate the size of the microgel in order to fit it with the desired microgel array spacing. This objective was achieved by introducing the surfactant CTAB during the polymerization process, thus allowing the production of microgels in the range between 753 and 162 nm, which were found to be optimally suited for microgel arrays printing with spacing of 1600 and 300 nm, respectively. We believe that this strategy improves on previous approaches for the generation of microgels and provides an efficient way for extending the range of spacing between microgel arrays.

The efficiency of cell adhesion and migration relies on a myriad of proteins, many of them being associated with the actin cytoskeleton, the functions of which must be orchestrated in time and space [70,74–76]. Tools for understanding cell adhesion and migration range from the generation of genetically modified cells, which lack or express mutated variants of actin-associated proteins, to the challenging of cells with chemically and topographically different substrates. For instance, cell adhesion can be directed using colloidal lithography to generate

**Table 3. Summary of the impact of microgels on the behavior of WT and KO Sertoli cells.**

| Sertoli[*] | films | | arrays | | arrays (5% x-linker)[$] | |
|---|---|---|---|---|---|---|
| | *WT* | *KO* | *WT* | *KO* | *WT* | *KO* |
| *Actin cytoskeleton organization* | random | random | aligned | aligned | n.a. | n.a. |
| *FA organization* | random | random | aligned | aligned | n.a. | n.a. |
| *Average speed* | – | - | ++ | + | n.a. | n.a. |
| *Directionality* | - | - | ++ | ++ | n.a. | n.a. |
| *FA assembly rate* | - | – | n | - | - | – |
| *FA disassembly rate* | - | – | n | n | n | – |
| *Zyxin kinetics* | n | n | - | + | n.a. | n.a. |

[*]All changes refer to control cells on glass substrata.

[$]Changes compared to 2% x-linker. (-): Decrease; (–): Strong decrease; (+): Increase; (++): Strong increase; n: No change; n.a.: Not analyzed.

adhesive islands with various shapes and sizes [47–49]. Cell migration, on the other hand, can be controlled by using several strategies including microgrooves [40,42], solution blow spinning fibers [10] or patterning of extracellular matrix proteins [41]. In this context, we have developed a tunable system based on surface-grafted microgels that efficiently modulates actin cytoskeletal architecture, cell adhesion and migration [50]. In the present study, the set of surface-grafted microgel arrays was expanded, including array spacing in the range between 300 and 1600 nm. Although, microgel arrays exerted a strong effect on the directionality of cell migration, we have identified a lower limit (300 nm) for array spacing which was less efficient in influencing directionality. As indicated by the AFM analysis, this was possibly due to the fact that the swelling of 300 nm microgel arrays in cell culture media caused adjacent arrays "to merge" resulting in a substrate (with no gaps between microgel lines) that resembled simple microgel films (see S6 Fig in S1 Appendix). One possible explanation for the residual cell alignment and directional motility is that even after swelling, 300 nm microgel arrays retain their parallel orientation, which is still and clearly sensed by the cells. In the microgel films, by contrast, the single microgel particle, though very close to each other, are homogenously distributed in all orientations thus providing no polarization cue to the cells. Furthermore, according to our previous work [50], we show that reduced cell migration on microgels (compared to glass control) was a general outcome regardless of the array spacing. As a general note, it could be argued that differences in cell behavior between microgel and glass controls could be, at least in part, due to different protein absorption on these substrata. Because this study was not aimed at addressing this issue, we cannot confirm nor rule out the effect of differential protein absorption on cell behavior. Certainly, future studies should take into account a potential difference in protein absorption between glass and microgel substrata.

Previous studies have shown that the spacing of topographic features plays a major role in the control of cell migration and directionality. For instance, a ridge/groove ratio of 1:3 promoted the greatest motility of osteoblasts on polystyrene substrates [77]. On polystyrene nanogrooves, osteoblasts exhibited the least directional migration when the topographic spacing was reduced to 150 nm [78]. Although the present findings and the earlier reported nanogroove experiments cannot be directly compared (due to the use of different materials and cell types), it is important to note that reducing the distance between adjacent topographic features (arrays in our study) decreased the directionality and speed of cell migration and represents, therefore, a general strategy for controlling cell motility. In this context, it is important to note that arrays' features such as spacing offer a very effective way to modulate cell motility and adhesion as clearly indicated by the significant differences between diverse arrays (see, for instance, S7A Table in S1 Appendix). This aspect will be very important for future optimization and applications of this type of substratum.

It is known that the speed of cell migration is correlated with focal adhesion dynamics [51,69,79]. In line with these studies, we have demonstrated that reduced focal adhesion dynamics in B16F1 and Sertoli cells on microgels corresponds to a reduction of the migratory speed of these cells. Therefore, it is likely that surface-grafted microgels modulate the rate of focal adhesion formation and disassembly resulting in a reduced cell migration. In spite of the different cell type and material used in the present study, our findings are consistent with the observation showing that smaller topographic poly(methyl methacrylate) hydrogel features promote faster NIH-3T3 cell migration and the formation of more dynamic focal adhesions [80]. According to this earlier study, in which focal adhesion dynamics was enhanced on a soft poly(methyl methacrylate) hydrogel, we found that focal adhesion dynamics and speed are reduced on more rigid microgel arrays. Thus, the right combination of surface properties is required for achieving optimal cell adhesion and adaptation to the substrate. Moreover, it is reasonable to suggest that surface-coated microgels can be exploited to efficiently modulate

cell adhesion and motility in the context of applications such as tissue engineering where these two biological events play a fundamental role.

## Conclusion

Cell adhesion and migration are fundamental for processes such as wound healing, and tissue regeneration where cell adhesion to, and migration on, a provisional extra cellular matrix is necessary for tissue formation. Since several cell types participate in these processes (e.g., fibroblasts and endothelial cells), a biomaterial should be designed in such a way that it can differentially and optimally support the adhesion and migration of all cell types involved. The design, fabrication and characterization of such biomaterials is very complex and beyond the scope of this study. Nevertheless, using simplified cellular systems, we have demonstrated that the variation of microgel array topographic and mechanical features can be efficiently used for the modulation of cell adhesion and motility. Our findings suggest that surface-grafted microgels could be potentially developed into a system capable of optimally supporting adhesion and migration of several cell types. We anticipate that the incorporation of chemical groups, variation of the degree of cross-linking and the fine tuning of substrate spacing could be used, alone or in combination, to confer on microgels the ability to (i) precisely modulate cell adhesion and migration of different cell types and (ii) develop implantable systems aimed at supporting and enhancing cell migration during, for instance, wound healing and tissue regeneration.

## Supporting information

**S1 Appendix. Supporting information includes the following supplementary data: Materials and methods, figures (Fig 1-6SUP), Tables (1–8) and References.**
(DOCX)

## Acknowledgments

We thank Gary Brook for critical reading of the manuscript and Gülcan Aydin for excellent technical assistance. We thank Sarah Staud and Oumaima Aiboudi for their help with the synthesis of the microgels.

## Author Contributions

**Conceptualization:** Andrij Pich, Antonio Sechi.

**Funding acquisition:** Martin Zenke, Andrij Pich.

**Investigation:** Janine Riegert, Alexander Töpel, Jana Schieren, Renee Coryn, Stella Dibenedetto, Dominik Braunmiller, Kamil Zajt, Carmen Schalla, Stephan Rütten, Martin Zenke, Andrij Pich, Antonio Sechi.

**Methodology:** Alexander Töpel, Andrij Pich, Antonio Sechi.

**Supervision:** Andrij Pich, Antonio Sechi.

**Writing – original draft:** Alexander Töpel, Antonio Sechi.

**Writing – review & editing:** Janine Riegert, Alexander Töpel, Jana Schieren, Renee Coryn, Stella Dibenedetto, Dominik Braunmiller, Kamil Zajt, Carmen Schalla, Stephan Rütten, Martin Zenke, Andrij Pich, Antonio Sechi.

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
