## [Decision Letter · Decision Letter 0]

29 Apr 2021

PONE-D-21-09835

Guiding cell adhesion and motility by modulating crosslinking and topographic
properties of microgel arrays

PLOS ONE

Dear Dr. Sechi,

Thank you for submitting your manuscript to PLOS ONE. After careful consideration, we
feel that it has merit but does not fully meet PLOS ONE’s publication criteria as it
currently stands. Therefore, we invite you to submit a revised version of the
manuscript that addresses the points raised during the review process.

Please submit your revised manuscript by Jun 13 2021 11:59PM. If you will need more
time than this to complete your revisions, please reply to this message or contact
the journal office at plosone@plos.org. When
you're ready to submit your revision, log on to https://www.editorialmanager.com/pone/ and select the 'Submissions
Needing Revision' folder to locate your manuscript file.

If you would like to make changes to your financial disclosure, please include your
updated statement in your cover letter. Guidelines for resubmitting your figure
files are available below the reviewer comments at the end of this letter.

We look forward to receiving your revised manuscript.

Kind regards,

Kerstin G. Blank

Academic Editor

PLOS ONE

Journal Requirements:

This work was partly supported by the Center for Chemical Polymer Technology (CPT),
which was supported by the EU and the federal state of North Rhine Westphalia (grant
EFRE 30 00883 02). AP thanks the financial support of the Deutsche
Forschungsgemeinschaft (DFG) of the Collaborative Research Center SFB 985
“Functional Microgels and Microgel Systems”. The funders have no role in study
design, data collection and analysis, decision to publish, or preparation of the
manuscript.

Additional Editor Comments:

When preparing your revised version, please pay special attention to the critical
comments of reviewer 3. In particular, comment on the concerns regarding possible
differences in cell adhesive protein adsorption, which could indeed affect the
observed differences between arrays and controls. Also, please discuss more
critically if differences between arrays are significant or if the presence of the
arrayed microgels is the primary effect. This will be important for future
optimization and possible applications.

In line with the PLOS data sharing policy, please also provide clear links connecting
the provided supplementary data with the figures (e.g. in the figure legends). If
not already done for all figures, please also provide all mean/median values used to
prepare the box plots as well as all values used to perform the statistical tests
(not only their outcome).

Reviewers' comments:

Reviewer's Responses to Questions

**Comments to the Author**

1. Is the manuscript technically sound, and do the data support the conclusions?

Reviewer #1: No

Reviewer #2: Yes

Reviewer #3: Yes

2. Has the statistical analysis been performed
appropriately and rigorously? 

Reviewer #1: Yes

Reviewer #2: Yes

Reviewer #3: Yes

3. Have the authors made all data underlying the
findings in their manuscript fully available?

Reviewer #1: Yes

Reviewer #2: Yes

Reviewer #3: Yes

4. Is the manuscript presented in an intelligible
fashion and written in standard English?

Reviewer #1: Yes

Reviewer #2: Yes

Reviewer #3: Yes

5. Review Comments to the Author

Reviewer #1: This manuscript describes the patterning of hydrogel microparticles and
the response of cell spreading to the corresponding interfaces. The impact of the
resulting patterns on the spreading and motility of Sertoli cells and Gas2L1 KO
cells was characterised. The dynamics of focal adhesions was then studied. The
authors then switch to the study of B16F1 cells for the study of the impact of
microgel size on cell adhesion, motility and adhesion turnover. Finally, the impact
of microgel mechanics was studied. Such phenomena and the impact of regulators of
actin assembly on sensing of such topographies are interesting, but important
aspects are not properly characterised or insufficiently detailed. The following
should be addressed prior to considering this manuscript any further:

1. Understanding factors regulating the sensing of nanotopography is important, but
it is not clear why select Gas2L1 specifically. Many other regulators of actin
assembly or FA regulators could have been picked up.

2. In addition, there are other platforms accessible to regulate nanoscale geometry
and topography. It is not clear why this specific microgel platform was selected and
how it allows to solve unanswered questions on the role of Gas2L1. Beyond the fact
that Gas2L1 regulates motility, a fact previously established, this study does not
demonstrate further roles for this player in topography sensing.

3. Similarly, it is not clear why the two cell models selected are particularly
appropriate to study the impact of microgel pattern topographies on cell adhesion.
It is also odd that some of the experiments are carried out only with B16F1 cells
when most of the study focuses on Sertoli cells. If the aim of the study is to mimic
topographical features present in the matrix, why not carry out all experiments in
the same cell type. The argument presented, that clearer trends could be observed
breaks down since no real trend correlating motility or FA turnover with microgel
topography (and size) is observed. Given that all other experiments were carried out
with Sertoli cells, studying the impact of microgel size on motility and FA turnover
seems important.

4. The differences observed in terms of cell motility and FA turnover are proposed to
result from differences in matrix. Clearly cells sense and align to the patterns
generated, but this was previously reported. However, differences observed between
glass, films and arrays (patterns) could arise from differences in cell adhesive
protein adsorption. I could not find details of any specific protein adsorption or
biofunctionalisation with peptides, therefore I assume that this was not controlled.
This aspect should be investigated for the different substrates studied (including
differences in microgel diameters and stiffness).

5. In Fig 6. FA assembly seems to be differentially regulated by Gas2L1 expression
when cultured on glass, but not when cultured on films or arrays. Trends (relative
rates of assembly/disassembly of FAs) are overall very similar for WT and KO lines.
This does not support the notion that Gas2L1 is mediating the sensing of the
microgel properties. It is not clear what properties are particularly targeted
either in this study (topography/size of microgels, chemistry, mechancis).

6. In Fig 7, the mobile fractions measured for the KO cells are lower than for WT, on
glass. However, in Fig 6, the assembly and disassembly rates of FA are faster. The
two observations seem to be contradicting.

7. In Fig 9, trends are proposed for the cell response to different microgels. E.g.
"their average speed and directionality were greatly reduced on 800, 1200 and 1600
nm arrays". Considering that there is no apparent statistical trend (the averages do
not even follow a clear trend when the microgel size was varied from 300 to 1600
nm), any notion that the microgel size (and associated topography?) is regulating
cell adhesion and motility should be removed from this manuscript. The only clear
difference observed is between glass/film and arrays. But again, this breaks down
when analysing the impact of substrate topography/patterning on FA
assembly/disassembly.

8. No statistical analysis results are reported in Fig 9.

9. What is the actual stiffness of the microgels used in Fig 10? This should be
measured.

10. Conclusions made in the discussion, specifically (but not only) "the migration of
B16F1 cells is inversely correlated with microgel array spacing" are not supported
by the data presented. This needs to be carefully rewritten. Similarly "In this
context, we have developed a (...) cell adhesion and migration". I do not see
evidence that the system presented allow to modulate actin cytoskeleton
architecture. There is no control in the phenotypes observed and no clear trends
when correlating spacing and migration/FA dynamics are observed. The cytoskeleton
architecture has not been systematically studied or characterised.

11. The conclusion that "the variation of microgel array topographic and mechanical
features can efficiently be used for the modulation of cell adhesion and motility"
is misleading. There is no real control achieved (which would be evidenced by trends
with microgel size for example). The only trends observed are the response of cells
to films and arrays compared to glass. But this could be the result of differences
in adhesive protein adsorption promoting differences in cell adhesion.

12. Fig. 7. It would be useful to show some of the normalised fluorescence
intensities prior to photobleaching in the traces presented.

13. The manuscript is sometimes vague and the language used could be more specific.
For example, "Focal adhesion dynamics is also modulated by microgels". What
properties of microgels? Or, "the kinetics of the focal adhesion protein zyxin is
decreased". The kinetics of which phenomenon? Presumably recruitment and
disassembly. Similarly, "whereas focal adhesion speed was clearly reduced". This is
presumably the FA formation rate, or the FA turnover.

Reviewer #2: Riegert et al investigated how surface bound microgel arrays affect cell
adhesion and migration. They controlled and varied the spacing of patterning and
degree of crosslinking and showed that cell adhesion and migration are indeed
affected. Greater spacing led to increased cell polarization and migration
directionality. The author further showed that this is directly correlated with the
focal adhesion dynamics. This is a very interesting study, and I find the article
generally well written and conclusions directly supported by the observation. I have
a few comments:

Fig 2 – are the second column a line cross-sectional profile or is it projected
vertically? In A, the periodicity is very clear, but seems to be diminished to noise
in B. Further, in C, the microgel is much narrower than the surface, but the line
profile in D seems to suggest they’re about the same width. The authors should
indicate where the cross-section is drawn on the first column that produced the
profile on the second column.

In the current design, both the gel with and the gap width seems to be increasing at
the same time. So, is it the microgel array width or their gap that has the stronger
effect on their influence to cell adhesion? For instance, keeping the gels at 300nm
width, but spacing hem 1600 nm apart, would the authors expect to see similar result
as gel/gap width = 300/300 or 1600/1600?

What is the mechanism behind the cells’ alignment in the direction of the pattern at
300nm (Fig 8A)? Given the authors stated that the surface swelling made it behave
like a connected sheet. How does the cell get the cues to polarize in this context
along the patterns. While the authors stated that the molecular mechanism is outside
the scope of the current study, can they provide some reasonable interpretations and
what they hypothesize might be the cause?

Fig 9 C, D doesn’t seem as significant as authors stated. For instance, in Fig 9C,
film, 800, 1600nm doesn’t seem to be different. Similarly, glass, 300 and 1200 do
not seem that different. How many independent experiments did the authors perform,
and how was the variation between experiments? Please include statistical test
between the necessary groups stated in the manuscript.

Reviewer #3: In this work, Sechi and coworkers developed surface-supported microgel
arrays featuring different spacing and elasticity and investigated the effect of
such topographic and mechanical cues on the adhesion and migration of different cell
types. Building up on previous research from their group, the authors optimized the
fabrication method of the biomaterial by varying the microgel size and crosslinking
degree to control the spacing between adjacent arrays as well as the materials
mechanical strength. After characterization of the synthesized microgels and the
printed microgel arrays, the authors used this platform to study the response of
cells (B16F1 and Sertoli cells) in terms of adhesion and migration in comparison to
cells cultured on glass and microgel films. Sertoli cells were found to elongate and
align their morphology, actin cytoskeleton and focal adhesions in respect to the
orientation of the arrays. Additionally, the rate of motility and directionality
were investigated, and a more pronounced impact of microgels on the migration of
Gas2L1 KO vs. wild-type Sertoli cells was found, while directionality was not
significantly impacted. Microgels topography and its stiffness was as well found as
effective means of modulating focal adhesion dynamism. Regarding B16F1 cells, their
migration average speed was reduced on microgel topography, and showed dependence on
the array spacing (with a lower limit of control found at 300 nm spacing),
reflecting differences in focal adhesion dynamics. The findings of this work are
useful to understand the influence of different design parameters of microgel
culture platforms on cell functionality. In a future perspective, these concepts
could guide the development of implantable scaffolds with controlled cell adhesion
and migration, to support wound healing and tissue engineering.

This is a technically sound investigation: the aim of the work is clearly formulated,
the experiments are competently performed, the analysis of different parameters on
cell response is exhaustive and the results are well presented. The findings are
interesting, the conclusions are supported by the data and the overall quality and
clarity of this manuscript are very good. Some experimental details are missing, and
some typos need correction, all aspects that can be easily implemented. I positively
support this article and recommend the acceptance for publication in PLOS One after
minor corrections noted (see below)

1- Experimental Section: some important missing details should be included:

- The authors mention the “microgel films” (Page 10) as one of the control substrates
to be compared to the “microgel array”. However, the preparation of the microgel
films is not described in the Exp. Section. Are they prepared similarly to the
microgel arrays, except that the printing part is skipped? Please, complete this
part for clarity.

- Page 6, line 120: “Microgels were subsequently purified by dialysis for several
days”. How many days were needed typically? Please, inform average purification
time.

- Page 8, formula to calculate the degree of microgel swelling: Please, define the
different terms or magnitudes of the equation at first mention. In connection to
that, how were the swelling degree and the volume phase transition temperature
measured? I suggest mentioning it briefly, even when they are shown in the Supp.
Info.

- Page 25, legend of Fig. 7, lines 554-555: “The thin lines above and below the thick
curves indicate the S.E.M.” Please, define “S.E.M”.

2- There are some typos to correct in the text:

- Page 4, line 90: “regulate” instead of “regulated”

- The authors vary between “cross linking” and “crosslinking” throughout the text.
Please, unify style. Same for “cross linker”, “crosslinker” and “cross-linker”.

- Supporting Information: “FTIR spectra” instead of “FITR spectra”.

- Page 25, line 565: “chose” instead of “chosen”.

- Page 31, line 715: “ratio” instead of “ration”.

3- My last comment is more a suggestion to the authors. With the investigation of
design parameters (spacing and stiffness) on diverse cell responses (morphology,
cytoskeleton and focal adhesion alignment, focal adhesion turnover, migration rate
and directionality, etc), across different substrates (glass, microgel array,
microgel films) and cell types (B16F1 cells, and wild-type vs Gas2L1 KO Sartoli
cells), at some point the reader may get lost with so many data. I believe that a
clear summary of the different findings at the end of the discussion section will be
very appreciated by the reader and will nicely wrap up the main findings. This could
be implemented, for example, in the form of a table that compares which factor has a
significant influence on a given response.

6. PLOS authors have the option to publish the peer
review history of their article (what does this mean?). If published, this will
include your full peer review and any attached files.

If you choose “no”, your identity will remain anonymous but your review may still be
made public.

**Do you want your identity to be public for this peer review?** For
information about this choice, including consent withdrawal, please see our
Privacy Policy.

Reviewer #1: No

Reviewer #2: No

Reviewer #3: No

---

## [Author Response · Author response to Decision Letter 0]

31 May 2021

see file Response to Reviwers

to Reviewers.docx
---

## [Decision Letter · Decision Letter 1]

23 Jun 2021

PONE-D-21-09835R1

Guiding cell adhesion and motility by modulating cross-linking and topographic
properties of microgel arrays

PLOS ONE

Dear Dr. Sechi,

Thank you for submitting your manuscript to PLOS ONE. After careful consideration, we
feel that it has merit but does not fully meet PLOS ONE’s publication criteria as it
currently stands. Therefore, we invite you to submit a revised version of the
manuscript that addresses the points raised during the review process.

Please submit your revised manuscript by Aug 07 2021 11:59PM. If you will need more
time than this to complete your revisions, please reply to this message or contact
the journal office at plosone@plos.org. When
you're ready to submit your revision, log on to https://www.editorialmanager.com/pone/ and select the 'Submissions
Needing Revision' folder to locate your manuscript file.

If you would like to make changes to your financial disclosure, please include your
updated statement in your cover letter. Guidelines for resubmitting your figure
files are available below the reviewer comments at the end of this letter.

We look forward to receiving your revised manuscript.

Kind regards,

Kerstin G. Blank

Academic Editor

PLOS ONE

Additional Editor Comments (if provided):

Reviewer 1 is now more positive about the manuscript. However, he stongly suggests to
adapt and rewrite some of the conclusions. I agree with this opinion. Please ensure
that the conclusions about the cellular response to the microgels are backed up by
the statistical analysis. In general, I strongly recommend to include the responses
to this reviewer in the manuscript or SI as the reader may have similar
questions.

Reviewers' comments:

Reviewer's Responses to Questions

**Comments to the Author**

1. If the authors have adequately addressed your comments raised in a previous round
of review and you feel that this manuscript is now acceptable for publication, you
may indicate that here to bypass the “Comments to the Author” section, enter your
conflict of interest statement in the “Confidential to Editor” section, and submit
your "Accept" recommendation.

Reviewer #1: (No Response)

Reviewer #2: All comments have been addressed

Reviewer #3: All comments have been addressed

2. Is the manuscript technically sound, and do the data
support the conclusions?

Reviewer #1: Partly

Reviewer #2: Yes

Reviewer #3: Yes

3. Has the statistical analysis been performed
appropriately and rigorously? 

Reviewer #1: Yes

Reviewer #2: Yes

Reviewer #3: N/A

4. Have the authors made all data underlying the
findings in their manuscript fully available?

Reviewer #1: Yes

Reviewer #2: Yes

Reviewer #3: Yes

5. Is the manuscript presented in an intelligible
fashion and written in standard English?

Reviewer #1: Yes

Reviewer #2: Yes

Reviewer #3: Yes

6. Review Comments to the Author

Reviewer #1: Reviewer: The authors have addressed some of the comments raised, but
left some of the most important comments regarding the validation of some of their
conclusions. Despite the replies made, the associated text and discussion were not
corrected appropriately. I still disagree with these conclusions and do not
recommend publication of this manuscript unless these aspects are fully and
appropriately addressed.

For clarity, I am attaching a pdf of my comments.

5. In Fig 6. FA assembly seems to be differentially regulated by Gas2L1
expression

when cultured on glass, but not when cultured on films or arrays. Trends (relative
rates

of assembly/disassembly of FAs) are overall very similar for WT and KO lines.
This

does not support the notion that Gas2L1 is mediating the sensing of the microgel

properties. It is not clear what properties are particularly targeted either in this
study

(topography/size of microgels, chemistry, mechanics).

In this part of the study, we have not analysed specific microgel properties but
the

impact of microgels on FA dynamics in control cells and cells lacking Gas2L1. We

clearly show that microgels are effective in regulating the motility and adhesion
of

Sertoli cells. Furthermore, FA assembly and disassembly rates are both reduced in

Gas2L1 KO cells compared to control cells on microgels arrays and films (see
suppl.

Table S4) indicating that Gas2L1 is somehow involved in sensing of microgel

substrata. Obviously, we do not know, at this stage, the molecular mechanisms for

this process that will be investigated in future studies.

R: I am not disputing the fact that microgels regulate the directionality of cell
motility. I am disputing the proposed role of Gas2L1 on this process. In Fig 5
(A-D), cells display high speed on the array compared to glass or film substrates,
whether they are WT or KO, although the speed of KO cells is overall reduced. The
trend in directionality is also the same. This is also mirrored by the identical
trends observed in Fig 6 (assembly and disassembly rates, and FA speed, are reduced
on films compared to arrays, both for WT and KO). Therefore, I conclude that,
although Gas2L1 has an impact on cell adhesion and motility, it does not regulate
the sensing of the topography. This needs to be corrected (and highlighted) in the
text, abstract, intro, discussion and conclusion.

Interestingly, such effect is more prominent in Sertoli cells harboring a knockout of
Gas2L1, a component of the cytoskeleton that mediates the interaction between
microtubules and microfilaments. Moreover, on microgel arrays, the kinetics of the
focal adhesion protein zyxin is decreased in wild-type and increased in Gas2L1 KO
Sertoli cells. Finally, increasing microgel cross-linking causes a stronger
reduction of focal adhesion turnover in Gas2L1 KO cells.

"we further reasoned that the present microgel system could be used to preferentially
modulate the migration of wild-type or Gas2L1 KO Sertoli cells. A correct hypothesis
would result in a clear difference in the rate of motility between the two Sertoli
cell lines with the Gas2L1 cells migrating faster than wild-type cells."

R: P21. L466. Again, I am not disputing that the WT and KO migrate at different rate,
they do. But they respond in a similar way to microgels and their topography,
compared to glass. Therefore, the microgel system is not preferentially modulating
the migration of WT or KO cells. Gas2L1 has an impact on migration, independent of
these substrates.

"A closer analysis of the data showed that the ratio between the average speeds of
Gas2L1 KO and wild-type cells was higher for cells on microgel films and microgel
arrays than for cells on glass coverslips (S3 Table), clearly showing a more
pronounced impact of microgels on the migration of Gas2L1 KO Sertoli cells."

"Conversely, in cells on microgel arrays, focal adhesion assembly was significantly
reduced only in Gas2L1 KO cells (Fig. 6A-B, D-E)."

"It is important to note that the ratio between the assembly rate of Gas2L1 KO and
wild-type cells was much smaller in cells on microgel films (S4 Table). The ratios
for focal adhesion disassembly rate and speed followed a similar trend (S4 Table).
Similar comparisons also showed that the focal adhesion size ratio was higher for
cells on microgel arrays, whereas focal adhesion life span ratio was higher for
cells on microgel films (S4 Table). Collectively, these findings demonstrate that
the surface-grafted microgels can be used as an effective system to modulate focal
adhesion dynamics in Sertoli cells and have a larger impact on Gas2L1 KO cells."

" Furthermore, the ratio between the mobile fractions of zyxin in wild-type and
Gas2L1 KO cells was increased in cells seeded on microgel arrays (S5 Table). Thus,
microgel arrays preferentially modulated zyxin kinetics in Gas2L1 KO cells, thus
serving as an effective tool for highlighting differences of focal adhesion behavior
between genotypically diverse cell types."

R: We noted the effort to characterise ratios between migration rates,
directionality, rates of assembly/disassembly etc. between KO and WT and compare
their ratios, but note that, as stated above, the overall trends remain unchanged
and there is no indication that the ratios reported in Tables S3-S5 demonstrate
quantitative differences in the response of KO and WT cells to patterns. Considering
the standard deviations, the min/max ratios (calculated from minimising/maximising
the corresponding values before calculating min/max ratios) are significantly
overlapping.

Overall, the role of Gas2L1 on cell migration is clear, but its role on sensing
topography of the microgels studied is not. This should be corrected in the
manuscript throughout and clearly stated.

6. In Fig 7, the mobile fractions measured for the KO cells are lower than for WT,
on

glass. However, in Fig 6, the assembly and disassembly rates of FA are faster.
The

two observations seem to be contradicting.

The mobile fraction of zyxin reflects its dynamic behaviour within focal
adhesions.

Such behaviour may or may not reflect the dynamic behaviour of focal adhesions
that

depends on the function of several proteins. Hence, in our opinion, the more
static

behaviour of zyxin in KO cells on glass does not necessarily represent a
“functional”

contradiction. In fact, it is possible that the behaviour of one single focal
adhesion

protein is affected in a certain way, whereas focal adhesions as a whole behave in
a

opposite way.

R: This still needs to be stated and discussed. There is no evidence for the
behaviour proposed by the authors. I do not recommend suggesting it without evidence
or without reference to appropriate literature.

7. In Fig 9, trends are proposed for the cell response to different microgels. E.g.
"their

average speed and directionality were greatly reduced on 800, 1200 and 1600 nm

arrays". Considering that there is no apparent statistical trend (the averages do
not

even follow a clear trend when the microgel size was varied from 300 to 1600 nm),

any notion that the microgel size (and associated topography?) is regulating cell

adhesion and motility should be removed from this manuscript. The only clear

difference observed is between glass/film and arrays. But again, this breaks down

when analysing the impact of substrate topography/patterning on FA

assembly/disassembly.

We thank the reviewer for the interesting comment on our interpretation of the
data.

We based our interpretation of the data and statements on the statistical analysis
(see

suppl. Tables S7 and S8). In our opinion there is a clear trend showing that the

average speed decreases as the microgel spacing increases.

R: I disagree. Fig 9 does not show stats, but 1600 nm is clearly above 1200 and 800
nm. Similarly, other processes quantified in Figure 9 clearly do not show trends.
Some stats should be included directly in this figure, to facilitate its
quantitative analysis.

Likewise, spacing greater than 300 nm consistently leads to higher directionality of
cell motility. We cannot, at present, precisely explain why cells on 1600 nm
microgel arrays regain part of their motility. However, we note that the 1600 nm
microgel arrays form a pyramidal structure characterised by one microgel sitting on
the top of two microgels. This

technical limitation arises by both the fact that we have used the largest
possible

PDMS stamp to print the 1600 nm arrays, and that physical limitations do not allow
to

generate larger microgels during the synthesis. Regardless of these current
technical

drawbacks, it is certainly important to investigate how cell behaviour changes
when

the distance between adjacent arrays increases.

R: clearly this calls for a modification of the conclusions. The microgel size does
not clearly modulate migration. Only 300 nm gels do.

10. Conclusions made in the discussion, specifically (but not only) "the migration
of

B16F1 cells is inversely correlated with microgel array spacing" are not supported
by

the data presented. This needs to be carefully rewritten. Similarly, "In this
context, we

have developed a (...) cell adhesion and migration". I do not see evidence that
the

system presented allow to modulate actin cytoskeleton architecture. There is no

control in the phenotypes observed and no clear trends when correlating spacing
and

migration/FA dynamics are observed. The cytoskeleton architecture has not been

systematically studied or characterised.

In our opinion, the statement “the migration of B16F1 cells is inversely correlated
with

microgel array spacing” is correct and supported by Fig. 9 (with the possible
exception

of the motility of cells on 1600 nm microgel arrays).

R: As noted above, there is no trend between migration speed, directionality and
assembly/disassembly rates in Figure 9. According to Table S7 cells migrate faster
on 1600 nm gels. The directionality is only significantly different for cells on 300
nm patterns. The assembly/disassembly rates follow a see-saw pattern, which does not
constitute a trend.

11. The conclusion that "the variation of microgel array topographic and
mechanical

features can efficiently be used for the modulation of cell adhesion and motility"
is

misleading. There is no real control achieved (which would be evidenced by trends

with microgel size for example). The only trends observed are the response of cells
to

films and arrays compared to glass. But this could be the result of differences
in

adhesive protein adsorption promoting differences in cell adhesion.

In our opinion, the statement "the variation of microgel array topographic and

mechanical features can efficiently be used for the modulation of cell adhesion
and

motility" highlighted by the reviewer is supported by solid experimental evidence.
As

to the topography, we have not only compared cell motility and adhesion on glass
to

the same biological processes on films and arrays, but also films to arrays.

Furthermore, we have changed arrays spacing (i.e., microgel topography) and
clearly

show that it can modulate cell adhesion and motility. Also in this case, the
comparison

was done with glass, but also with films and between pairs of different array
spacings

(see statistical analysis in the suppl. data). Regarding the mechanical feature
(i.e.,

content of cross-linker), we clearly show that focal adhesion turnover is clearly

modulated by this microgel feature in both WT and Gas2L1 KO cells.

R: As noted above, this still requires revising. This study presents some interesting
results, but I dispute some of its conclusions and this should not be discarded.

12. Fig. 7. It would be useful to show some of the normalised fluorescence
intensities

prior to photobleaching in the traces presented.

As the reviewer certainly knows, the normalised intensities prior to photobleaching
are

equal to 1 and will be visualised as curves parallel to the X axis for all the
conditions

(i.e., glass, films and arrays). In our opinion, the information will not add any
relevant

detail to the figure but introducing it will cause the curves to be squeezed
together

(because the Y axis will include a range of values up to 1 or more) thus making
any

difference difficult to appreciate.

R: The value of presenting normalised fluorescence intensities directly prior to
bleaching (for a few tens of s is to clearly show whether some gradual
photobleaching of the systems was observed simply during imaging.

R: Statistical Analysis.

Experiments should all be carried out at least in triplicates, rather than duplicates
and triplicates.

Reviewer #2: (No Response)

Reviewer #3: The authors have addressed my suggestions satisfactorily. I recommend
publication since the revised manuscript now meets the journal standards.

7. PLOS authors have the option to publish the peer
review history of their article (what does this mean?). If published, this will
include your full peer review and any attached files.

If you choose “no”, your identity will remain anonymous but your review may still be
made public.

**Do you want your identity to be public for this peer review?** For
information about this choice, including consent withdrawal, please see our
Privacy Policy.

Reviewer #1: No

Reviewer #2: **Yes: **Isaac T.S. Li

Reviewer #3: No

---

## [Author Response · Author response to Decision Letter 1]

4 Aug 2021

see file Response to Reviewers

to Reviewers.docx
---

## [Decision Letter · Decision Letter 2]

3 Sep 2021

Guiding cell adhesion and motility by modulating cross-linking and topographic
properties of microgel arrays

PONE-D-21-09835R2

Dear Dr. Sechi,

We’re pleased to inform you that your manuscript has been judged scientifically
suitable for publication and will be formally accepted for publication once it meets
all outstanding technical requirements.

Kind regards,

Kerstin G. Blank

Academic Editor

PLOS ONE

Additional Editor Comments (optional):

As you are interested in publishing the Peer Review History, I am happy to provide
the necessary information. If the submission is accepted for publication, you will
be invited to opt-in to publish the Peer Review History of the manuscript, using a
form in our Editorial Manager submission system. If you should not receive that
query from us after editorial acceptance, please let us know.

I personally think the manuscript will benefit from publication of the Peer Review
History, as the discussion between you and the reviewers contains useful additional
information for the reader.

Reviewers' comments:

Reviewer's Responses to Questions

**Comments to the Author**

1. If the authors have adequately addressed your comments raised in a previous round
of review and you feel that this manuscript is now acceptable for publication, you
may indicate that here to bypass the “Comments to the Author” section, enter your
conflict of interest statement in the “Confidential to Editor” section, and submit
your "Accept" recommendation.

Reviewer #1: All comments have been addressed

2. Is the manuscript technically sound, and do the data
support the conclusions?

Reviewer #1: Partly

3. Has the statistical analysis been performed
appropriately and rigorously? 

Reviewer #1: Yes

4. Have the authors made all data underlying the
findings in their manuscript fully available?

Reviewer #1: Yes

5. Is the manuscript presented in an intelligible
fashion and written in standard English?

Reviewer #1: Yes

6. Review Comments to the Author

Reviewer #1: Most of the comments made have been addressed. Some of the issues raised
could be better addressed and I would still dispute some of the conclusions, but
this would require further investigation and I understand it would delay
publication.

7. PLOS authors have the option to publish the peer
review history of their article (what does this mean?). If published, this will
include your full peer review and any attached files.

If you choose “no”, your identity will remain anonymous but your review may still be
made public.

**Do you want your identity to be public for this peer review?** For
information about this choice, including consent withdrawal, please see our
Privacy Policy.

Reviewer #1: No

---

## [Editor Report · Acceptance letter]

14 Sep 2021

PONE-D-21-09835R2 

Guiding cell adhesion and motility by modulating cross-linking and topographic
properties of microgel arrays 

Dear Dr. Sechi:

I'm pleased to inform you that your manuscript has been deemed suitable for
publication in PLOS ONE. Congratulations! Your manuscript is now with our production
department. 

Kind regards, 

on behalf of

Dr. Kerstin G. Blank 

Academic Editor

PLOS ONE